# Structure of the human heparan-α-glucosaminide *N*-acetyltransferase (HGSNAT)

**Vikas Navratna[1,2], Arvind Kumar[3], Jaimin K Rana[1,2], Shyamal Mosalaganti[1,2,4]\***

[1]Life Sciences Institute, University of Michigan, Ann Arbor, United States; [2]Department of Cell and Developmental Biology, University of Michigan, Ann Arbor, United States; [3]Thermo Fisher Scientific, Waltham, United States; [4]Department of Biophysics, College of Literature, Science and the Arts, University of Michigan, Ann Arbor, United States

**\*For correspondence:**
mosalaga@umich.edu

**Competing interest:** The authors declare that no competing interests exist.

**Abstract** Degradation of heparan sulfate (HS), a glycosaminoglycan (GAG) comprised of repeating units of *N*-acetylglucosamine and glucuronic acid, begins in the cytosol and is completed in the lysosomes. Acetylation of the terminal non-reducing amino group of α-D-glucosamine of HS is essential for its complete breakdown into monosaccharides and free sulfate. Heparan-α-glucosaminide *N*-acetyltransferase (HGSNAT), a resident of the lysosomal membrane, catalyzes this essential acetylation reaction by accepting and transferring the acetyl group from cytosolic acetyl-CoA to terminal α-D-glucosamine of HS in the lysosomal lumen. Mutation-induced dysfunction in HGSNAT causes abnormal accumulation of HS within the lysosomes and leads to an autosomal recessive neurodegenerative lysosomal storage disorder called mucopolysaccharidosis IIIC (MPS IIIC). There are no approved drugs or treatment strategies to cure or manage the symptoms of, MPS IIIC. Here, we use cryo-electron microscopy (cryo-EM) to determine a high-resolution structure of the HGSNAT-acetyl-CoA complex, the first step in the HGSNAT-catalyzed acetyltransferase reaction. In addition, we map the known MPS IIIC mutations onto the structure and elucidate the molecular basis for mutation-induced HGSNAT dysfunction.

## eLife assessment

This **important** study presents the structure of human heparan-alpha-glucosaminide N-acetyltransferase (HGSNAT) in the acetyl-CoA bound state, providing the first description of the architecture of this family of integral membrane enzymes, and revealing the mode of acetyl-CoA binding. The structural work is **convincing**, with a high resolution and isotropic single-particle cryoEM map and an atomic model that is well-justified by the density map, with strong density for the acetyl-CoA ligand. However, experimental support for the molecular mechanism of the HS acetylation reaction and the impact of disease-causing mutations is **incomplete**. This work will be of interest to biochemists and structural biologists studying the structure and function of integral membrane enzymes, as well as those interested in genetic diseases resulting from mutations in this family of enzymes, such as mucopolysaccharidosis IIIC (MPS III-C).

## Introduction

Degradation of the extracellular matrix proteoglycans begins in the cytosol, where they are proteolyzed to GAGs such as HS, chondroitin sulfate, dermatan sulfate, and keratan sulfate. The proteolyzed GAGs are further degraded to monosaccharides and free sulfate residues within the lysosomes

in GAG-specific multi-enzyme degradation pathways. Dysfunction, often inherited, of the enzymes involved in GAG degradation causes abnormal accumulation of the partial degradation products within the lysosomes, resulting in mucopolysaccharidoses (MPSs), a collective term for the group of rare autosomal recessive lysosomal storage disorders characterized by the accumulation of partially degraded GAGs (*Coutinho et al., 2012*; *Huizing and Gahl, 2020*; *Klein et al., 1978*; *Platt et al., 2018*; *Ruivo et al., 2009*). HS is a GAG made of repeating units of *N*-acetylglucosamine and glucuronic acid, and impairment of enzymes in the HS degradation pathway causes mucopolysaccharidosis III (MPS III), or Sanfilippo's syndrome that is characterized by abnormal storage of HS degradation pathway intermediates within the lysosomes in all organs and excretion of these intermediates in the urine. The onset of the symptoms of MPS III, which include organomegaly, abnormal joint mobility, compromised cardiac function, degeneration of vision and hearing, and dementia, often begins in juveniles and ends in premature death. Currently, no therapies for MPS III are available (*Coutinho et al., 2012*; *Huizing and Gahl, 2020*; *McBride and Flanigan, 2021*; *Pshezhetsky et al., 2018*).

Depending on the dysfunctional enzyme, MPS III is classified into four subtypes – MPS IIIA-IIID. Of these four MPS III subtypes, MPS IIIC (prevalence rate of 1 in 1.4 million) is caused by the dysfunction of heparan-α-glucosaminide *N*-acetyltransferase (HGSNAT, EC 2.3.1.78). HGSNAT is the only enzyme of the GAG pathway that is not a hydrolase. It catalyzes the only known biosynthetic reaction of the GAG degradation pathway within the lysosome, that is, the acetyl-CoA (ACO) mediated *N*-acetylation of the terminal non-reducing amino group of α-D-glucosamine (*Fan et al., 2006*; *Hrebícek et al., 2006*; *Huizing and Gahl, 2020*; *Klein et al., 1978*; *Nagel et al., 2019*; *Pshezhetsky et al., 2018*). Acetylation of the terminal α-D-glucosamine group is essential for subsequent HS degradation within the lysosomes. HGSNAT is not homologous to any known proteins, including acetyl-CoA binding proteins, *N*-acetyltransferases, and other lysosomal proteins. HGSNAT mRNA has two translation start sites (M1 and M29), yielding two functionally active isoforms (73 kDa and 70 kDa) of HGSNAT, both targeted to the lysosomal membrane (*Durand et al., 2010*; *Fan et al., 2011*). In this study, we use isoform 2 of HGSNAT, which is produced as a 635 amino acid protein containing a 30 amino acid N-terminal signal peptide, a ~110 amino acid long luminal domain, and 11 transmembrane helices (TMs). HGSNAT is believed to be expressed in the cell as an immature precursor protein that localizes as an *N*-glycosylated dimer on the lysosomal membrane (*Durand et al., 2010*; *Fan et al., 2011*). The localization of HGSNAT happens via the adaptor protein-mediated pathway, aided by the lysosomal targeting motifs *[DE]-XXXL[LI]* ([204]ETDRLI[209]) and *YXXØ* ([624]YILYRKK[630]) present towards the N- and C-terminus of HGSNAT, respectively (*Rudnik and Damme, 2021*; *Schwake et al., 2013*). Deleting the C-terminal lysosomal sorting signal in HGSNAT retains the protein in the plasma membrane (*Bonifacino and Traub, 2003*; *Durand et al., 2010*). Once targeted to the lysosomal membrane, HGSNAT, like a few other lysosomal membrane proteins, is proteolyzed by unidentified acid proteases in the lysosomal lumen into two unequal fragments - a smaller luminal N-terminal α-HGSNAT and a larger transmembrane C-terminal β-HGSNAT that continue to co-localize despite the proteolysis (*Durand et al., 2010*; *Fan et al., 2011*; *Rudnik and Damme, 2021*; *Steenhuis et al., 2012*). Upon proteolytic maturation, it is believed that HGSNAT is assembled as a hetero oligomer of α- & β-HGSNAT chains (*Durand et al., 2010*; *Feldhammer et al., 2009b*). However, the essentiality of proteolytic cleavage and oligomerization for HGSNAT activity remains debated as endoplasmic reticulum (ER) retained monomeric HGSNAT was also shown to be active (*Durand et al., 2010*; *Fan et al., 2011*).

So far, over 70 unique mutations in the *HGSNAT (TMEM76)* gene have been identified. These mutations span the entire sequence and include deletions, nonsense mutations, splice-site variants, and silent and missense mutations (*Canals et al., 2011*; *Fan et al., 2006*; *Fedele and Hopwood, 2010*; *Feldhammer et al., 2009a*; *Feldhammer et al., 2009b*; *Hrebícek et al., 2006*; *Huizing and Gahl, 2020*). The majority of the mutations that cause MPS IIIC are missense mutations that result in HGSNAT folding and localization defects, making them ideal targets for pharmacochaperone therapy. Site-specific inhibitors of HGSNAT activity have been explored as agents to rescue the misfolded conformation of certain missense variants, thereby restoring the partial *N*-acetyltransferase activity (*Coutinho et al., 2012*; *Fedele and Hopwood, 2010*; *Feldhammer et al., 2009b*; *Pan et al., 2022*). Despite its clinical relevance, the structure of HGSNAT and the mechanism of *N*-acetyltransferase activity are poorly understood. Here, using single-particle cryo-EM, we report the high-resolution structure of full-length HGSNAT in a complex with acetyl-CoA. This is the first structure of a member of the transmembrane acyl transferase (TmAT) superfamily (classes 9.B.97 and 9.B.169 in the Transporter

Classification Database (TCDB)) (*Saier et al., 2021*). The HGSNAT-acetyl-CoA complex structure presented here provides a high-resolution snapshot of the first step in HGSNAT catalyzed acetyl-transferase reaction, and reveals critical cofactor binding amino acids in the HGSNAT active site. In addition, our structure also provides molecular insights into the impact of MPS IIIC-causing mutations on acetyl-CoA binding and the overall architecture of the protein.

## Results

### Purification of dimeric HGSNAT

Two groups have previously reported purification of HGSNAT in varying oligomeric forms (*Durand et al., 2010*; *Fan et al., 2011*; *Feldhammer et al., 2009b*). The first group used non-ionic surfactants, NP-40 and Triton X-100, in their purification experiments and observed dimers and hexamers of HGSNAT. The PEG-based headgroup in these detergents is relatively bulky and yields micelles of variable sizes (45–100 kDa) and has been known to cause sample heterogeneity in eukaryotic membrane proteins (*Orwick-Rydmark et al., 2016*). The second group purified monomeric HGSNAT using non-ionic detergent DDM, which forms a relatively uniform, albeit large, micelle (98 kDa). However, the purification process involved two overnight incubation steps – the first in 1% and the second in 0.2% DDM. Both groups used transient transfection to express the protein in COS-7 or HeLa cells, respectively. All these factors could impact the monodispersity of purified HGSNAT. To identify the ideal conditions for expression and purification of monodisperse HGSNAT, we transfected HEK293 GnTI⁻ cells with plasmids expressing either N-terminal or C-terminal GFP-fusion of HGSNAT and monitored GFP fluorescence in detergent-solubilized HGSNAT lysates by fluorescence-detection size-exclusion chromatography (FSEC) (*Kawate and Gouaux, 2006*). We noticed that the position of GFP did not alter the expression of HGSNAT (*Figure 1—figure supplement 1A*). However, the HGSNAT dimer model predicted using ColabFold suggested that the C-termini of the protomers lie at the dimer interface (*Mirdita et al., 2022*). Thus, for large-scale production of HGSNAT for structural studies, we expressed N-terminal StrepII-tag-GFP-HGSNAT fusion. Large quantities of GFP-HGSNAT fusion were produced using baculovirus-mediated transduction of HEK293 GnTI⁻ cells (*Goehring et al., 2014*). We found that reducing the temperature of the suspension cell culture to 32 °C at about 8–10 hr after transduction, or transient transfection, resulted in a better yield than continuing to grow cells at 37 °C (*Figure 1—figure supplement 1B*). To identify the ideal conditions for the purification of HGSNAT, we screened various detergents. In most of the detergent conditions we tested, HGSNAT was predominantly dimeric (*Figure 1—figure supplement 1C–H*). Then, we performed a single-point thermal melt test of solubilized cell lysates to compare the relative stability of HGSNAT in different detergents. The rationale is that heating should exacerbate the instability of the protein in a said condition. HGSNAT appeared most stable in digitonin (*Figure 1—figure supplement 1I–L*). Based on our FSEC analysis of relative thermal stability and homogeneity, we purified HGSNAT in digitonin. We observed a dimer in our chromatography experiments and a band corresponding to a monomer on SDS-PAGE (*Figure 1—figure supplement 1M–O*).

### Architecture of HGSNAT

We solved the structure of HGSNAT in complex with acetyl-CoA (HGSNAT-ACO complex) to a global resolution of 3.26 Å, with the transmembrane domain (TMD) being better resolved than the luminal domain (LD) (*Table 1*, *Figure 1—figure supplement 2*, and *Figure 1—figure supplement 3A–C*). This is the first experimentally determined structure of a member of the protein family (TmAT). Our structure reveals that HGSNAT is a dimer where the protomers are related to each other by twofold rotational symmetry, and the C2 axis of rotation is perpendicular to the plane of the membrane (*Figure 1A–C*). Each polypeptide of HGSNAT has an N-terminal LD followed by 11 TMs that comprise the TMD, which is ensconced in the lysosomal membrane. We could model all the secondary structure elements of the protein unambiguously except the first 48 amino acids including the N-terminal signal peptide and the cytoplasmic loop 1 (CL1; L183-L236). The regions that are poorly resolved are the β-turn that connects β7 and β8 (N140-E148) the C-terminal half of TM1 (V176-F181) (*Figure 1A*, and *Figure 1—figure supplement 3D and E*). The C-terminus of the protein lies at the dimer interface but is unlikely to be directly involved in dimerization as the C-termini of the protomers lie ~25 Å away from each other, pointing towards the central acetyl-CoA binding site (ACOS) (*Figure 1B and C*).

**Table 1.** Cryo-EM data collection, processing, and validation statistics.

| | EMDB-41620 and PDB-8TU9 |
|---|---|
| *Data collection and processing* | |
| Magnification | 105,000 x |
| Voltage (kV) | 300 |
| Data collection mode | Super-resolution |
| Electron exposure (e–/Å$^2$) | 50 |
| Defocus range (µm) | –1.0 to –2.5 |
| Physical Pixel size (Å) | 0.848 |
| Symmetry imposed | C2 |
| Initial particle images (no.) | 3,325,732 |
| Final particle images (no.) | 57,739 |
| Map resolution (unmasked, Å) at FSC 0.143 | 3.7 |
| Map resolution (masked, Å) at FSC 0.143 | 3.3 |
| Map resolution range (Local resolution) | 2.5–4.5 |
| *Refinement* | |
| Map sharpening B factor (Å$^2$) | –30 |
| Model composition | |
| *Chains* | 4 |
| *Atoms* | 8284 (Hydrogens: 0) |
| *Residues* | Protein: 1066 Nucleotide: 0 |
| *Water* | 0 |
| *Ligands* | ACO: 2 |
| Bonds (RMSD) | |
| *Length (Å) (#>4sigma)* | 0.008 (0) |
| *Angles (°) (#>4sigma)* | 1.480 (28) |
| MolProbity score | 1.25 |
| Clash score | 2.32 |
| Ramachandran plot (%) | |
| *Outliers* | 0.00 |
| *Allowed* | 3.59 |
| *Favored* | 96.41 |
| Rama-Z (Ramachandran plot Z-score, RMSD) | |
| *whole (N=1058)* | –0.40 (0.24) |
| *helix (N=486)* | 1.03 (0.21) |
| *sheet (N=114)* | 1.69 (0.51) |
| *loop (N=458)* | –2.35 (0.23) |
| Rotamer outliers (%) | 0.00 |
| Cβ outliers (%) | 0.00 |
| Peptide plane (%) | |
| *Cis proline/general* | 0.0/0.0 |

*Table 1 continued on next page*

*Table 1 continued*

| | EMDB-41620 and PDB-8TU9 |
|---|---|
| *Twisted proline/general* | 0.0/0.0 |
| Cα BLAM outliers (%) | 4.38 |
| ADP (B-factors) | |
| *Iso/Aniso (#)* | 8284/0 |
| *Protein (min/max/mean)* | 10.47/125.39/28.58 |
| *Ligand (min/max/mean)* | 16.71/16.71/16.71 |
| *Model vs. Data* | |
| CC (mask) | 0.78 |
| CC (box) | 0.63 |
| CC (peaks) | 0.58 |
| CC (volume) | 0.72 |
| Mean CC for ligands | 0.77 |

## Transmembrane domain (TMD)

The TMD of the HGSNAT protomer comprises of a central 4+4 fold, where the TMs 2–5 are related to TMs 6–9 by a twofold rotational pseudo-symmetry with the axis of rotation being perpendicular to the plane of the membrane (*Figure 1D–F*). TMs 2–5, along with TM10, form a 'catalytic core' and enclose the ACOS accessible via the cytosol and lumen along the dimer interface. ACOSs of the protomers lie on either side of the dimer interface axis. TMs 6–9, along with TM11, form a 'scaffold domain' separated from the dimer interface by the catalytic core (*Figure 1E*). The third luminal loop (LL3) connecting TM6 and TM7 forms a lid that limits the access to the cavity between the catalytic core and the scaffold domain from the luminal side (*Figure 1A, C and D*). The TMD region comprising TM2-TM11 of HGSNAT is predicted to be evolutionarily conserved across HGSNATs from other kingdoms (*Figure 2—figure supplement 1A*). Owing to the unique architecture we have named this novel fold as the transmembrane N-acetyltransferase (TNAT) fold. Although the resolution for the C-terminal half of TM1 is poor, it is sufficient to position the helix separately from the rest of the TMD (*Figure 1A*). TM1 is connected to TM2 by ~50 amino acid long CL1 (*Figure 1D*). LL1-LL3 and LL5 form the boundaries for the luminal entrance of ACOS. The C-terminus, CL2, and a part of CL1 form the boundaries of the cytosolic entrance of ACOS. The ACOS in our structure is relatively more accessible from the luminal side than the cytosolic side, as the nucleoside head group of the bound acetyl-CoA blocks the cytosolic entrance of ACOS (*Figure 1C*). Thus, the HGSNAT-ACO complex is in a conformation where the active site is readily accessible for the binding of the second substrate from the lysosomal lumen.

## Luminal domain (LD)

The LDs of the protomers lie diagonally opposite to each other ~45 Å away from the dimer interface and are not involved in dimerization (*Figure 1B*). Located between the N-terminal signal peptide and the TM1, LD is a~110 amino acids long β-sandwich made of two beta sheets of four strands each where strands β1, β4, β7, & β8 form the mixed β-sheet on top (*Figure 1A–E*, blue), and the strands β2, β3, β5, & β6 arrange as bottom anti-parallel β-sheet (*Figure 1A–E*, gray). The mixed β-sheet is arranged such that the order of the strands is β4-β1-β7-β8, with β1 & β7 being parallel. β8 of LD is connected to TM1 (*Figures 1B, D and 2A*). LD has two predicted disulfide bonds – one in the β-turn that connects β2 and β3 (C76-C79), and the other between the strands β6 and β8 (C123-C151) holding the two sheets together. The resolution of the LD domain in our structure allows us to model only one (C76-C79) of these unambiguously (*Figure 2B*). HGSNAT is produced as a pro-protein that is believed to get proteolyzed into two fragments, HGSNAT-α and HGSNAT-β, of unequal sizes, which remain together (*Durand et al., 2010*; *Fan et al., 2011*). It is unclear if HGSNAT-α is made of just LD or LD and TM1, and if HGSNAT-β is made of all TMs or only TMs2-11 (*Figure 2A* and *Figure 2—figure supplement 1*). While our structure has relatively poor local resolution at the predicted protease sites,

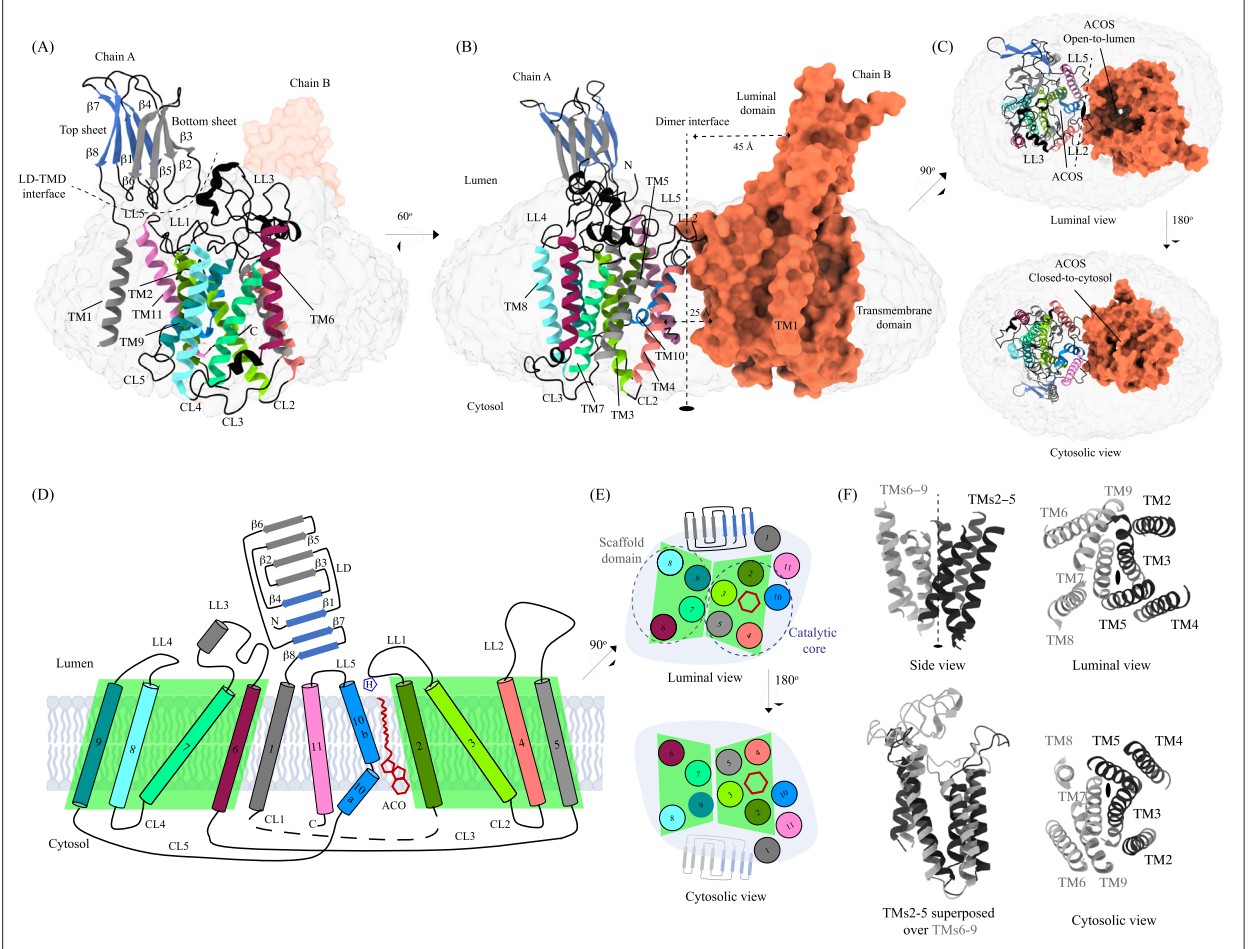

**Figure 1.** Structure of heparan-α-glucosaminide N-acetyltransferase (HGSNAT). Panels (**A**) and (**B**) show two different orientations of the HGSNAT dimer that highlight (dashed lines) the LD-TMD interface and dimer interface, respectively. Micelle is displayed in gray. Chain A is displayed as a cartoon and chain B as an orange surface. All the luminal loops (LLs), cytosolic loops (CLs), and the loops that connect β-sheets are shown in black. The top and bottom sheets in the luminal domain (LD) are colored blue and gray, respectively. The twofold rotation axis is displayed as a dashed line with an ellipsoid. (**C**) Luminal (top) and cytosolic (bottom) views of the protein. The surface representation of chain B suggests that the acetyl-CoA binding site (ACOS) is more accessible from the luminal side (top) than the cytosolic side (bottom). (**D**) 2D topology of HGSNAT and YeiB family. The helices and strands in the topology are colored similarly to the 3D structure. Transmembrane helices (TMs) 2–5 and 6–9 form two bundles (4+4), highlighted by green parallelograms, that are related to each other by a twofold rotation parallel to the plane of the membrane. TMs 1, 10, and 11 do not seem involved in this internal symmetry, with TM10 being bent in the plane of the membrane into two halves TM10a and TM10b. The relative position of bound ACO and active site H269 of LL1 are indicated. (**E**) Luminal (top) and cytosolic (bottom) views of the protein topology. TMs 2–5 and TM10 enclose ACOS (red hexagon) and are referred to as catalytic core (blue dashed oval). TMs 6–9 will be referred to as scaffold domain (gray dashed oval). (**F**) 4+4 bundle formed by TMs 2–5 (black) and TMs 6–9 (gray) are related by a twofold rotation. The last sub-panel (bottom left) shows a superposition of TMs 2–5 on TMs 6–9.

The online version of this article includes the following source data and figure supplement(s) for figure 1:

**Figure supplement 1.** Purification of heparan-α-glucosaminide N-acetyltransferase (HGSNAT).

**Figure supplement 1—source data 1.** Uncropped and labeled gel for *Figure 1—figure supplement 1M*.

**Figure supplement 1—source data 2.** Raw unedited gel for *Figure 1—figure supplement 1M*.

**Figure supplement 2.** Cryo-electron microscopy (Cryo-EM) data processing workflow.

**Figure supplement 3.** Cryo-electron microscopy (Cryo-EM) data quality, reconstruction, and model building.

making it difficult to map them unambiguously, the FSEC nor the SDS-PAGE analyses of the purified protein indicate that the recombinant HGSNAT is not proteolyzed (*Figure 1—figure supplement 1*). We believe the relatively low local resolution of LD is because of the flexibility introduced by GFP fused to the N-term of HGSNAT.

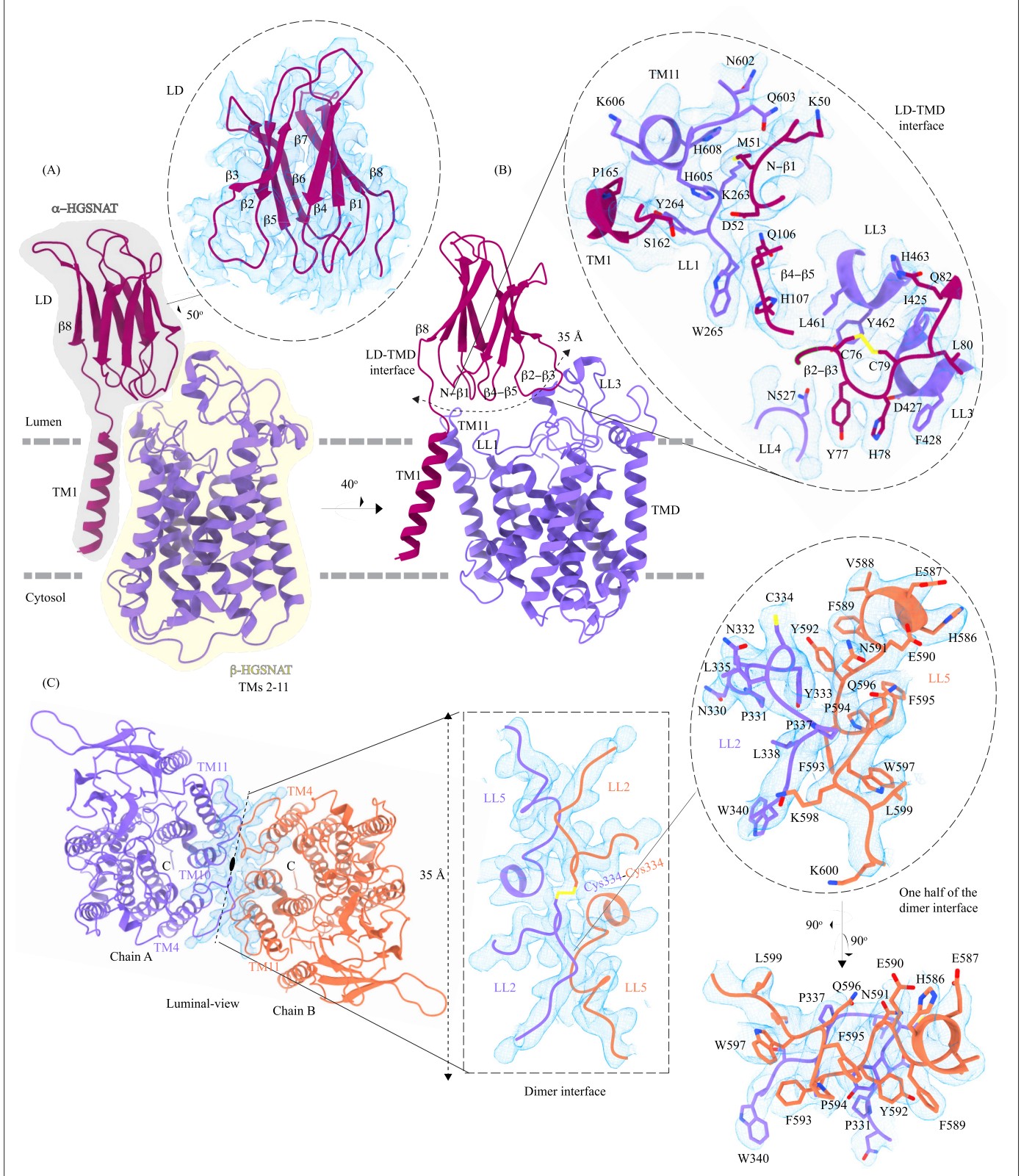

**Figure 2.** Domain organization, and LD-TMD and dimer interfaces of heparan-α-glucosaminide N-acetyltransferase (HGSNAT). (**A**) HGSNAT is predicted to be proteolyzed into two chains of unequal size - α-HGSNAT (dark magenta cartoon, gray shaded area) and β-HGSNAT (purple cartoon, yellow shaded area). The site for proteolysis remains debated. Based on our structure and prediction of HGSNAT structures from other kingdoms (**Figure 2—figure supplement 1**), we have represented α- and β-HGSNAT fragments as shown in panel A. The inset (dashed oval) shows the luminal

*Figure 2 continued on next page*

*Figure 2 continued*

domain (dark magenta) fit to cryo-electron microscopy (cryo-EM) density (blue; display level 0.21 of the composite map in ChimeraX) (*Figure 1—figure supplement 3*). The lysosomal membrane is shown as a dashed gray line. (**B**) LD-TMD interface is highlighted (dashed line). Inset highlights the residues that interact at the LD-TMD interface, and cryo-EM density for the same (blue; display level 0.25 of the 3.26 Å C2 refined map in ChimeraX). C76-C79 disulfide of β2-β3 turns is shown as yellow sticks, while the residue sidechains are colored the same as their secondary structure elements, with heteroatoms highlighted. (**C**) Luminal view of the protein with dimer interface highlighted (dashed line). Inset (dashed rectangle) highlights LL2 and LL5 that line the dimer interface, and the C334-C334 inter-chain disulfide (yellow) between the chains A (purple) and B (orange). The dashed oval inset shows one-half of the dimer interface with LL2 and LL5 of chains A and B, respectively, contributing other hydrophobic interactions that stabilize the dimer interface. The cryo-EM density in panel C is displayed as blue mesh (display level 0.22 of the C2 refine map in ChimeraX).

The online version of this article includes the following figure supplement(s) for figure 2:

**Figure supplement 1.** Homologs of heparan-α-glucosaminide N-acetyltransferase (HGSNAT).

**Figure supplement 2.** Interactions at LD-TMD and dimer interface.

**Figure supplement 3.** Lipids and detergent in the structure.

## LD-TMD interface

LD interacts with TMD at multiple sites via an extensive interaction network that spans ~35 Å, comprised of a salt bridge (D52-H605), hydrogen bonds, and dipole-dipole and hydrophobic interactions (*Figure 2B* and *Figure 2—figure supplement 2A*). The N-terminus of TM11 interacts with the N-terminus of β1 and LL1. LL1 also interacts with the N-terminus of TM1 and the β4-β5 turn. The β2-β3 turn, including the C76-C79 disulfide, interacts extensively with LL3 and is a part of LL4. The β2-β3 turn-LL3-LL4 interaction is also stabilized by a 3π-network formed by the stacking of Y77, H78, & F428 side chains. The hydrogen bond network that stabilizes the LD-TMD interaction is formed between the amino acids M51, H78, and S162 of LD with H605 and Q603, D427 and F428, and Y264 of the TMD, respectively (*Figure 2B* and *Figure 2—figure supplement 2A*). The LD-TMD interface is separated from the dimer interface by the central catalytic core (*Figure 2B and C*).

## Dimer interface

The dimer interface is spread across ~35 Å towards the luminal side of the protein, perpendicular to the C2 rotation axis (*Figure 2C*). Although the TMs 4, 10, and 11 from both protomers lie on either side of the dimer interface, they do not directly interact to stabilize the dimer interface. The primary mediator of dimerization is the disulfide between C334s within the LL2s of each protomer (*Figure 2C*, rectangle inset). In addition to the C334-C334 disulfide, the dimer interface is also stabilized by an extensive π-π interaction network formed between the aromatic residues of LL2 (332-340) of one protomer and the aromatic residues of LL5 (588-598) of the other protomer (*Figure 2C* and *Figure 2—figure supplement 2B*). Notably, the aromatic bulky side chains at the interface are buried in the membrane, and the hydrophilic residues face the lumen (*Figure 2C*, oval inset). Y333, L338, and S339 of LL2 from one protomer form a series of hydrogen bonds with F593 and K598 of LL5 of the other protomer, which adds to the stabilization of the dimer interface (*Figure 2C* and *Figure 2—figure supplement 2B*). In all the detergents we tested, HGSNAT eluted as a dimer, a testimony to the extensive side-chain interaction network. The twofold rotational symmetry between the protomers juxtaposes the ACOSs of protomers connecting them with each other on the cytosolic side (*Figures 1 and 2C*). This interconnected space is partitioned by lipids, preventing the diffusion of ligands from one active site to the other within a dimer (*Figure 2—figure supplement 3*). We generated two HGSNAT mutants of residues at the dimer interface – C334A and F593A. C334 from each protomer is linked in a disulfide bond, and F593 forms hydrogen bonds with multiple residues from the opposite protomer (*Figure 2C* and *Figure 2—figure supplement 2B*). In both these mutants, the overall expression of HGSNAT was not reduced. In the case of F593A, the stability of the dimer also remained unaffected (*Figure 4—figure supplement 1A,H,I and N*). While both these mutants still expressed predominantly as dimers even in the presence of 1% digitonin, the C334A mutant eluted as a monomer upon heat treatment (*Figure 4—figure supplement 1E,H,I and K*). As C334A breaks a covalent disulfide link between the protomers, in the absence of this disulfide, the extensive side-chain interaction network stabilizes the dimer interface. Heating destabilizes these interactions, resulting in a monomeric C334A.

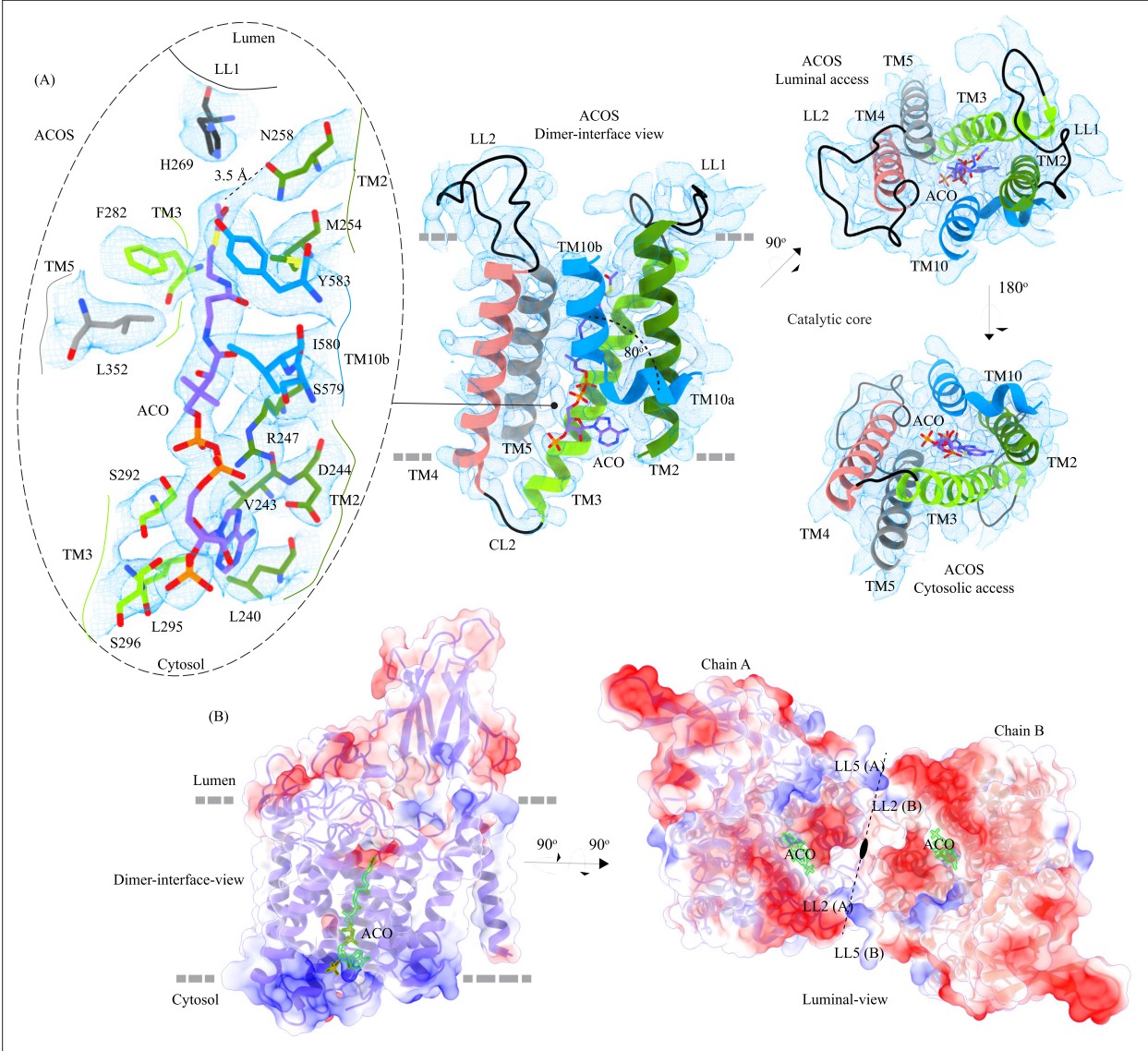

**Figure 3.** Acetyl-CoA binding site (ACOS). (**A**) Catalytic core (chain A) of heparan-α-glucosaminide N-acetyltransferase (HGSNAT) comprised of transmembrane helices (TMs) 2–5 and TM 10. Luminal loops (LLs) and cytosolic loops (CLs) are shown in black, and the helices are colored as in *Figure 1*. Acetyl-CoA (ACO) is colored (purple), the same as chain A in *Figure 2* with heteroatoms highlighted. The inset (dashed oval) shows acetyl-CoA binding site (ACOS) and highlights the amino acids of HGSNAT that interact with ACO. The amino acids are colored the same as the corresponding TMs, with heteroatoms highlighted. Cryo-electron microscopy (Cryo-EM) density for ACOS is displayed as blue mesh (display level 0.3 of the 3.26 Å C2 refine map in ChimeraX). ACO could be modeled into the densities at chain A and B ACOSs with a mean correlation coefficient (CC) of 0.77. The nucleoside headgroup of ACO plugs in the cytosolic access of ACOS, and the luminal access seems relatively more accessible. (**B**) Electrostatic potential and surface charge distribution of HGSNAT, with the surface display colored based on the potential contoured from −10 kT (red) to +10 kT (blue). ACO bound at the ACOS is highlighted in golden yellow. Luminal and cytosolic sides of the protein show a conspicuous polarity. The lysosomal membrane is shown as a dashed gray line in both sub-panels.

The online version of this article includes the following figure supplement(s) for figure 3:

**Figure supplement 1.** Ligand binding sites of heparan-α-glucosaminide N-acetyltransferase (HGSNAT).

## Acetyl-CoA binding site (ACOS)

We used MOLEonline to predict the presence of a~75 Å pore within the TMD, that begins at the cytosolic side and extends to the luminal side (*Pravda et al., 2018*; *Figure 3—figure supplement 1A*). The pore predicted by MOLEonline is lined by highly conserved residues of the TMs 2–5 and TM 10 (*Figure 3* and *Figure 3—figure supplement 1C*). We note that the helix TM10 is bent by ~80° such that the luminal-side 2/3rd (TM10b) of the helix is parallel to TM2 and the cytosolic-side

1/3rd (TM10a) of the helix protrudes between TM2 and TM11 (*Figure 3A*). The bending of TM10 is aided by P575 and G576, residues are known to induce helix breaks and kinks (*Javadpour et al., 1999*; *Ulmschneider and Sansom, 2001*). We find a density in our cryo-EM map overlapping the MOLEonline prediction in both protomers, spanning the entire length of TMD, that allowed us to unambiguously model a single acetyl-CoA, such that the 3', 5'-ADP nucleoside head group is towards the cytosol and the acetyl group is hydrogen bonded (~3.5 Å) to N258 of TM2 that is located at the luminal entrance of ACOS. On the cytosolic side, the orientation of TM10a between TM2 and TM11 seemed to have allowed the positioning of the 3', 5'-ADP nucleoside head group in the space created by bending away of TM10a from the central axis of the catalytic core. The catalytic H269, which is predicted to be acetylated during the acetyltransferase reaction, is ~4.5 Å away from the acetyl group of acetyl-CoA, nestled within the negative charge on the luminal opening, and is ergonomically positioned to perform acetyltransferase reaction in the presence of the acetyl group acceptor (*Figure 3A*). The binding pocket, like the rest of the protein, shows a polarity of charge. As we move from the cytosolic side to the luminal side, the charge changes from positive to neutral to negative (*Figure 3B* and *Figure 3—figure supplement 1A and D*). The cytosolic opening of the site is comprised of basic and nonpolar amino acids, making it an ideal pocket for binding the 3', 5'-ADP head group. The pantothenate group of acetyl-CoA is supported by a network of nonpolar amino acids at the center of the binding pocket (*Figure 3—figure supplement 1A and D*). A series of conserved salt bridges stabilize the cytosolic and luminal entrances of the pore. The luminal entrance is lined with salt bridges between the residues H269, D279, R344, D469, E471, H586, and E587. The cytosolic entrance is lined with salt bridges between the residues R239, D244, R247, R317, E363, K491, and K634 (*Figure 3—figure supplement 1C and D*). A portion of CL1 towards the TM2 and C-terminus of the protein also seems to be a part of the cytosolic entrance of the ACOS.

## Conservation and homology

Sequence conservation between HGSNAT and other known acetyl-CoA binding proteins or transferases is poor. There are no structural homologs of HGSNAT, and a search within the database of known structures using the HGSNAT TMD by the Dali server resulted in hits with sequence identity of <17% across alignments of <25% sequence length (*Holm et al., 2023*). Even the search with just the luminal domain, which appeared to be a classic two-sheet β-sandwich, yields hits with <15% identity over alignments of at least 70% sequence lengths (*Supplementary file 1*, *Figure 2—figure supplement 1C and D*). We used ModelAngelo, a machine-learning-based de novo automatic model-building algorithm to build the initial HGSNAT model into the cryo-EM density (*Jamali et al., 2023*). The model built by ModelAngelo into the experimental data superposed well with the HGSNAT model as predicted by AlphaFold with a Cα RMSD of ~1.7 Å over 533 amino acids (*Figure 2—figure supplement 1A*; *Jumper et al., 2021*; *Varadi et al., 2022*). However, unlike the AlphaFold model of Isoform 1, our expression construct lacks the extended signal peptide (N-terminal 28 amino acids). In addition, we do not observe any density in model CL1 that connects TM1 and TM2. HGSNATs of representatives from archaea, bacteria, and plants suggest that LD and TM1 are absent in other kingdoms. These homologs superpose onto the TMD part of HGSNAT, especially TM2-11 with Cα RMSDs of 1–1.4 Å over 350 amino acids (*Figure 2—figure supplement 1*). TmAT superfamily of membrane proteins consists of two subfamilies of integral membrane proteins consisting of 8–12 TMs – the acyltransferase-3/acetyl-CoA transporter (ATAT) family (TCDB: 9.B.97) and the integral membrane acetyltransferase (YeiB) family (TCDB: 9.B.169). HGSNAT belongs to the YeiB family. A meaningful sequence alignment could not be generated between the members of these two families. ATAT family of transporters seems to have a single extra-membranous domain, like HGSNAT (*Figure 2—figure supplement 1B*). However, based on AlphaFold prediction, this extra-membranous domain seems to be a α-β-α sandwich with a single 5-stranded β-sheet sandwiched between two helical domains. Incidentally, a bent helix near the predicted acetyl-CoA binding site is also observed in the ATAT family, suggesting that this could be a conserved feature amongst the members of the TmAT superfamily (*Newman et al., 2023*). We used ConSurf to estimate sequence conservation in HGSNAT based on an automatic sequence alignment algorithm (*Ashkenazy et al., 2016*). It appears that the LD and TM1 regions have poor sequence conservation compared to the rest of the protein (*Figure 4A*).

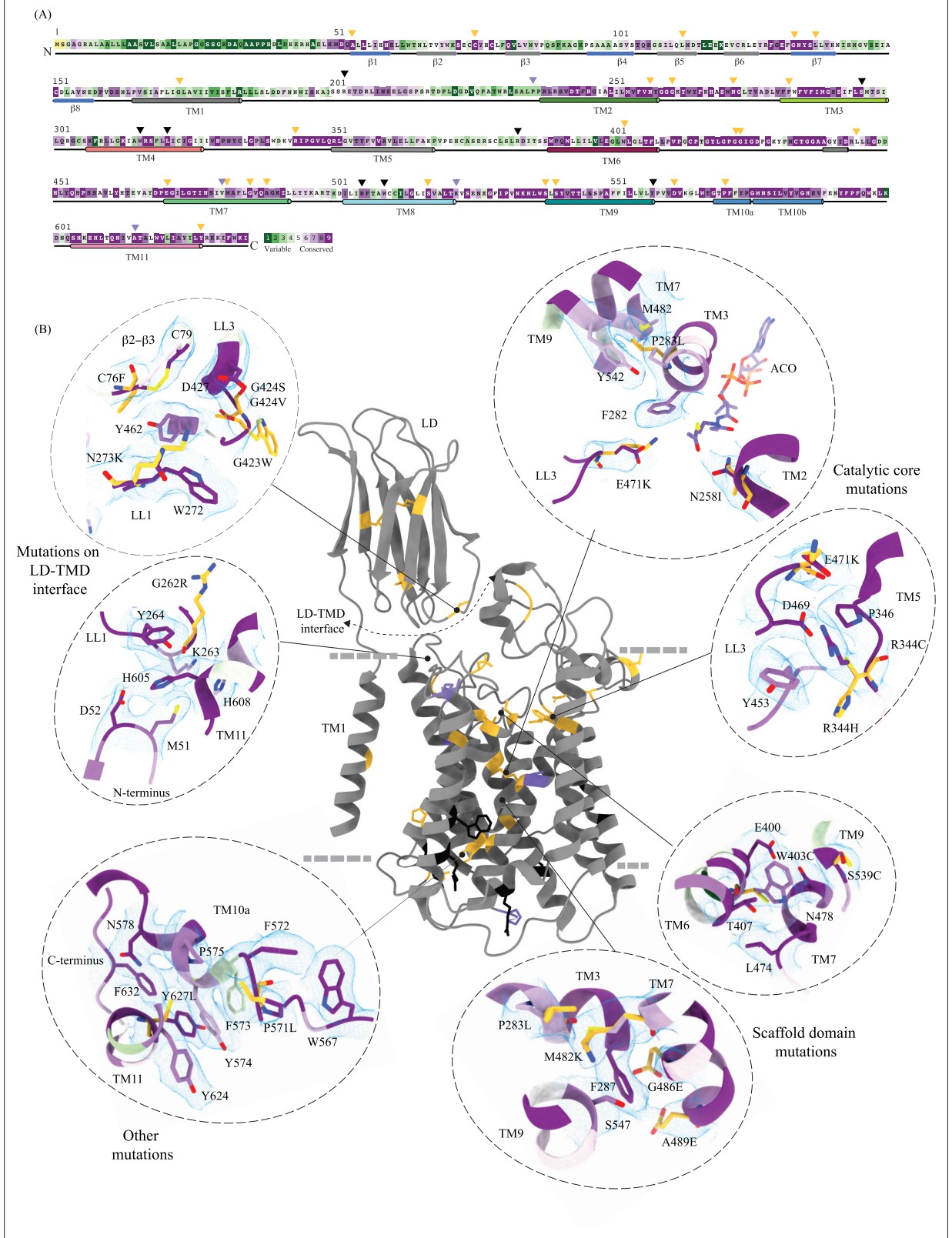

**Figure 4.** Molecular basis for mucopolysaccharidosis IIIC (MPS) IIIC mutation-induced dysfunction. (**A**) Evolutionary sequence conservation of heparan-α-glucosaminide N-acetyltransferase (HGSNAT). Amino acids are color-coded according to the conservation scores generated by the ConSurf web server using a Clustal multiple sequence alignment of homologs identified by PSI-BLAST (**Ashkenazy et al., 2016**). The positions of the mutations - missense (orange), nonsense (black), and polymorphisms (purple) – are indicated on the sequence by triangles. (**B**) MPS IIIC-causing mutations

*Figure 4 continued on next page*

*Figure 4 continued*

mapped on the HGSNAT structure. The color coding of the positions is the same as in panel A. Some of the missense mutants are highlighted in the insets (dashed ovals). We grouped them based on their position within the protein – LD-TMD interface, catalytic core, scaffold domain, and other C-terminal mutations. The insets show the 3D environment of the mutant sites on the wild-type HGSNAT color coded as per their evolutionary sequence conservation scores, and the potential disturbance to it caused by the mutation (orange side chains). The coordinates for mutant side chains were generated based on wild-type HGSNAT structure as input in FoldX webserver (*Schymkowitz et al., 2005*).

The online version of this article includes the following figure supplement(s) for figure 4:

**Figure supplement 1.** Expression and stability of heparan-α-glucosaminide N-acetyltransferase (HGSNAT) mutants.

## The basis for mutation-induced dysfunction and destabilization of HGSNAT

We mapped the known clinical mutations (missense, nonsense, and polymorphisms) of HGSNAT onto its three-dimensional structure (*Canals et al., 2011*; *Fan et al., 2006*; *Fedele and Hopwood, 2010*; *Feldhammer et al., 2009a*; *Feldhammer et al., 2009b*; *Hrebícek et al., 2006*; *Huizing and Gahl, 2020*). Almost all the mutations fall within the conserved regions on HGSNAT (*Figure 4A*). The nonsense mutations seem to be located more on the cytosolic side, and the missense mutations seem to be populated on the luminal side of HGSNAT (*Figure 4B*). We estimated the extent of destabilization or stabilization introduced by these variants on HGSNAT structure by FoldX, a force field algorithm to evaluate the effect of mutations on the stability and dynamics of proteins (*Supplementary file 2*; *Schymkowitz et al., 2005*). We classified these mutations into four groups based on their position on the structure and found that most destabilizing mutations appear to be concentrated near the LD-TMD interface (*Figure 4* and *Supplementary file 2*). Our structure provides the basis for destabilization or dysfunction induced by missense mutations in HGSNAT.

### Catalytic core mutations

Five missense mutations - E471K (LL3), R344C/R344H (LL2), P283L (TM3), and N258I (TM2) - have been identified in the residues that line the ACOS (*Figures 4B and 3A*, and *Figure 3—figure supplement 1C*). E471 is close to the acetyl group of acetyl-CoA, and acidic to basic side chain substitution in an E471K mutation could impact the binding affinity of acetyl-CoA. N258 side chain can act as both hydrogen acceptor or donor and is hydrogen bonded to acetyl-CoA in the structure. N258I mutation could directly impact the binding of acetyl-CoA and acetyltransferase activity. R344 forms salt bridges with E469 and E471 that stabilize the luminal entrance of ACOS. The mutations of R344C/R344H could affect the integrity of the luminal entrance of ACOS. In addition to being involved in acetyl-CoA binding, all these three positions – E471, R344, and N258 – are also highly conserved (*Figure 4A and B*). P283 is not directly involved in the binding of acetyl-CoA. However, the residues flanking P283 – V281, F282, F285, I288, and M289 – all interact with the pantothenate group of acetyl-CoA (*Figure 3A* and *Figure 3—figure supplement 1D*). Proline residues define helix conformation, and mutation of a relatively conserved proline to an aliphatic leucine could alter the TM3 conformation on the luminal side and thus affect acetyl-CoA binding. FoldX prediction indicates that P283L is the most destabilizing of all ACOS mutations perhaps because it impacts the TM3 conformation. In contrast, other catalytic core mutants only show a potential to impact the binding of acetyl-CoA. We generated N258I and R344H mutants to test the effect of these substitutions on the expression and stability of HGSNAT. We noticed that these substitutions did not reduce the overall expression of HGSNAT (*Figure 4—figure supplement 1A*). R344H mutant, upon solubilization in 1% digitonin, showed slightly enhanced aggregation but the overall stability of R344H mutant was not altered drastically, compared to WT HGSNAT (*Figure 4—figure supplement 1F and I*, and 1 L). Although the N258I mutant showed expression comparable to WT HGSNAT, as evident by total fluorescence measurements of solubilized cell lysates, we could not observe any peak for N258I nor free GFP in FSEC (*Figure 4—figure supplement 1A and C*). We hypothesize that N258I substitution directly affected the substrate binding and stability. As a result, upon solubilization followed by ultracentrifugation, the unstable protein got pelleted out of the solution (*Figure 4B* and *Figure 4—figure supplement 1C*).

## Mutations at the LD-TMD interface

C76F, G262R, N273K, G423W, and G424S/G424V are all missense mutations that are on the LD-TMD interface, and our FoldX-based analysis indicates these mutations to be the most destabilizing among the ones that we listed. N273K is predicted to be least destabilizing by FoldX, as all these mutations, except N273K, result in charge reversal and drastic change in the side chain size, leading to steric clashes and breakdown of existing interactions (*Figures 4B and 2B*). Although none of these missense positions at the LD-TMD interface, except C76, are directly involved in LD-TMD interface contacts, the drastic side chain changes in the vicinity of the residues directly involved are expected to destabilize the interaction (*Figure 2* and *Figure 2—figure supplement 2*). For example, the glycine residues (G262, G423, and G424) lie within pockets lined by aromatic side chains containing amino acids, and the substitution of such a residue with a bulky side chain will cause a steric clash, destabilizing those pockets of interaction (*Figure 4B*). C76F mutation reduces the expression of the protein, suggesting that LD-TMD interaction is essential for proper folding and stability of HGSNAT (*Figure 4—figure supplement 1A and B*).

## Scaffold domain mutations

W403C, M482K, G486E, A489E, S518F, S539C, and S541L are all mutations that occur in the scaffold domain (TM6-TM9) of HGSNAT. FoldX predicts these mutations to be only mildly destabilizing. For example, the change in the charge and size of S539C or W403C mutation is not drastic enough to destabilize TM6 or TM9 helices. Even in the cases where there are large substitutions, for example, G486E or A489E, the substituted side chains face away from the core of the protein and are involved in minimal interactions or clashes (*Figure 4B*). Corroborating the FoldX prediction, we noticed that W403C was mildly destabilizing. While the overall expression of the W403C mutant was not affected, the thermal stability was reduced compared to WT HGSNAT (*Figure 4—figure supplement 1A,G,I and M*).

## Other mutations

Mutations in the LD domain, such as L113P, G133A, and L137P, seem to face the lumen and are involved in minimal interactions. Amongst the mutations that we analyzed by FoldX, none of them were on the dimer interface. However, two mutations, P571L and Y627C, towards the cytosolic ends of TM10 and TM11 seem closer to the dimer interface on the cytosolic side. However, they are not directly involved in dimerization. These ring-side chain amino acids are part of an elaborate π-π-network stabilizing TM10a and the C-terminus in positions that make space for the nucleoside head group at the ACOS (*Figures 4B and 3A*). Drastic mutations in this region could impact acetyl-CoA binding and thereby the conformational stability of the catalytic core.

## Mechanism of acetyltransferase reaction

HGSNAT, in this report, was purified at pH 7.5. During the purification process, we did not add acetyl-CoA to the buffer. However, we were able to confirm the presence of endogenously bound acetyl-CoA in our cryo-EM sample by LC-MS (*Figure 5—figure supplement 1*). It has been demonstrated, previously, that acetyl-CoA binding happens on the cytosolic side and is optimal around pH 7.0–8.0, and the acetyl transfer activity happens on the luminal side and is optimal at pH 5.5–6.0. The $K_m$ of *N*-acetyltransferase reaction in the presence of acetyl-CoA as the substrate is in the 0.2–0.6 mM range at pH 5–5.8, and 2.5–20 µM range at pH 7.0, suggesting that HGSNAT has a high affinity for acetyl-CoA at pH $\geq$7.0 (*Bame and Rome, 1985*; *Bame and Rome, 1986a*; *Meikle et al., 1995*). The transfer of the acetyl group from acetyl-CoA to the terminal non-reducing amino group of α-D-glucosamine is believed to be catalyzed by H269 located on the luminal entrance of ACOS (*Figure 3A* and *Figure 3—figure supplement 1*). It is unclear if H269 gets acetylated and forms a stable acetylated intermediate in the process of acetyl transfer. However, for the H269 to be acetylated or for it to transfer the acetyl group to its putative acceptor, protonation of the imidazole sidechain amine group is necessary. The catalytic H269 is primarily uncharged at pH 7.5, and the acetyl-CoA remains an acetate ion. The carboxamide group of N258 can act as both an acceptor and donor in a hydrogen bond. In our structure, ACOS is more accessible from the luminal side than the cytosolic entrance. We note that the acetyl-CoA is hydrogen-bonded to N258 (*Figure 3A* and *Figure 3—figure supplement 1D*). Based on these observations, we conclude that the structure we have is of an acetyl-CoA primed

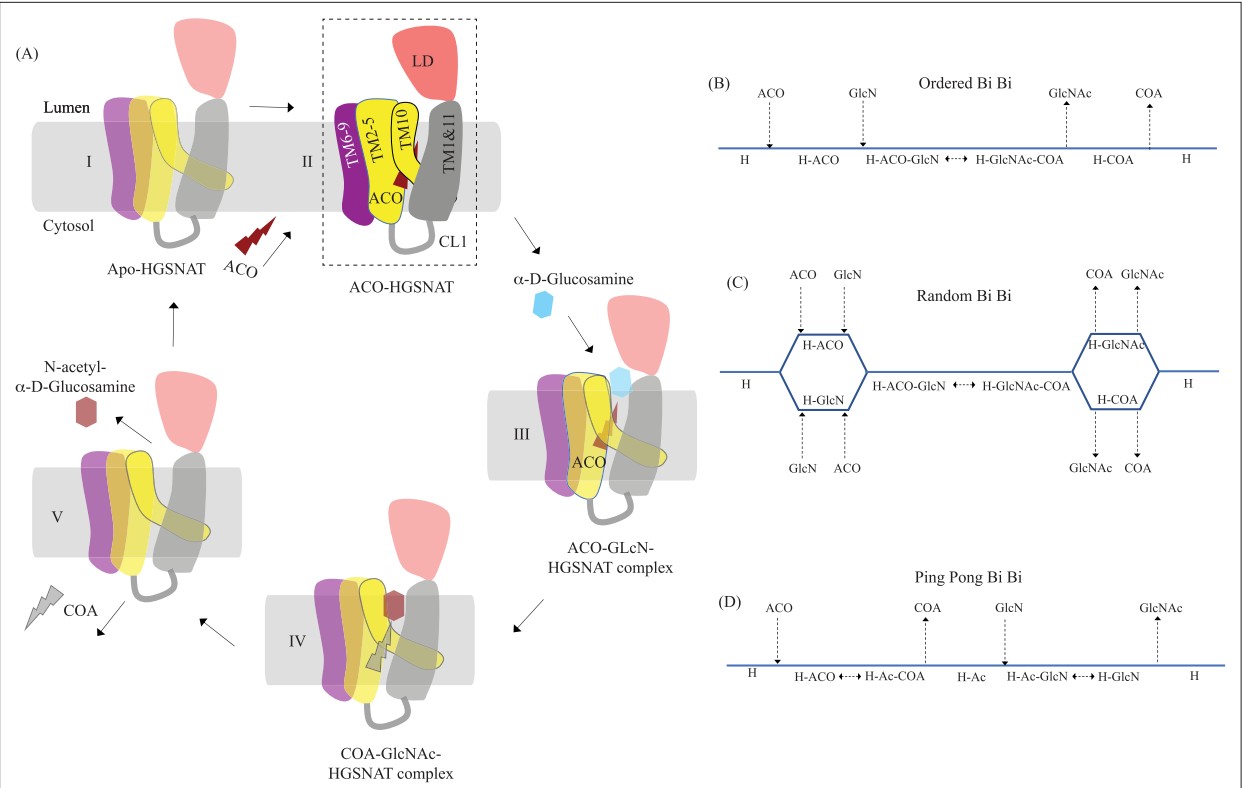

**Figure 5.** Proposed mechanism of acetyl transfer by heparan-α-glucosaminide N-acetyltransferase (HGSNAT). (**A**) HGSNAT (I) catalyzes a bisubstrate reaction of transferring acetyl group from cytosolic acetyl-CoA (ACO, red lightning) to terminal non-reducingα-D-Glucosamine (GlcN, blue hexagon) of luminal heparan sulfate (III and IV). After the acetyl group transfer, COA (gray lightning) and acetylated glucosamine (GlcNAc, red hexagon) are believed to be released to cytosol and lumen, respectively (V). Depending on the order of binding and release of substrates and products, enzyme-catalyzed bisubstrate reactions could either be sequential reactions (**B and C**) or ping pong reactions (**D**). The mechanism of reaction catalyzed by HGSNAT has been a longstanding debate. We believe that the acetyl-CoA bound HGSNAT structure presented in this work (II, dashed box) is in a cofactor primed conformation which could proceed by any of the bisubstrate reaction mechanisms shown in **B–D**. The function of luminal domain (LD) is unclear, and we believe it plays essential role in recognition of substrate and its positioning at the active site.

The online version of this article includes the following figure supplement(s) for figure 5:

**Figure supplement 1.** LC-MS analysis of purified heparan-α-glucosaminide N-acetyltransferase (HGSNAT).

HGSNAT waiting for the protonation of H269 and the availability of an acetyl group acceptor to carry out acetyltransferase reaction (*Figure 5A*). Because the cytosolic pH is favorable for acetyl-CoA binding and the protein we purified endogenously pulled down acetyl-CoA with it, we believe this is the most stable conformation of HGSNAT in the absence of an acetyl group acceptor. Furthermore, in our FSEC-based analysis, we noticed that H269A substitution did neither alter the expression nor stability of HGSNAT (*Figure 4—figure supplement 1A,D and J*). However, the N258I mutation completely destabilized HGSNAT (*Figure 4—figure supplement 1A and C*). We believe N258I instability to be a consequence of lack of substrate binding at the catalytic core, as we see a greater role for N258 in acetyl-CoA binding as opposed to H269 as per our structure (*Figure 3A* and *Figure 3—figure supplement 1*).

## Discussion

Direct involvement of acetyl-CoA in the heparan sulfate degradation was shown in the early 1980 s, by incubating the extracted intact lysosomes with radiolabeled acetyl-CoA and by subsequently monitoring the incorporation of radiolabeled acetate into lysosomal HS (*Rome and Crain, 1981*; *Rome et al., 1983*). Using purified lysosomal membranes, Rome and colleagues showed that the acetyl-CoA-dependent *N*-acetyltransferase activity, which is required for acetylation of HS, resides on the lysosomal membrane. In these studies, they also suggested that the HS acetylation reaction occurs

in two steps - acetylation of the lysosomal membranes and transfer of acetyl from the membranes to HS. By performing acetylation and acetyltransferase reactions at different pH, and in the presence of different amino acid modification reagents, they showed that the acetylation process is optimal at pH 7, acetyltransferase activity is optimal at pH 5.5, and the amino acid on the lysosomal membrane that gets acetylated is a histidine. They proposed that the *N*-acetyltransferase activity follows a Ping Pong Bi Bi mechanism of reaction, where acetyl-CoA from the cytosol acetylates the active site histidine of the *N*-acetyltransferase located on the lysosomal membrane, inducing a conformational change in the transmembrane enzyme that enables the access of active site histidine from the luminal side (*Figure 5D*). Due to the change in pH of the active site cavity, the acetyl-histidine interaction is destabilized, and the acetyl group is transferred to the α-D-glucosaminide on the luminal side (*Bame and Rome, 1985*; *Bame and Rome, 1986a*; *Bame and Rome, 1986b*). Pshezhetsky and colleagues showed that acetylation of lysosomal membranes expressing *N*-acetyltransferase activity occurred even in the absence of the acetyl group acceptor, suggesting the formation of an acetylated enzyme intermediate (*Ausseil et al., 2006*; *Durand et al., 2010*). Here, we purified HGSNAT at a pH favorable for acetyl-CoA binding, but not acetyl transfer reaction. In our structure, we find that the HGSNAT ACOS is more accessible via the lumen as opposed to the cytosol and that the His269 of LL1 is not acetylated. Instead, we find an acetyl-CoA molecule in the ACOS, hydrogen bonded to N258 of TM2, a residue within a close vicinity (~5 Å) of H269 (*Figure 3* and *Figure 3—figure supplement 1*). We believe N258 holds onto acetyl-CoA and aids H269 in the acetyltransferase reaction, at low pH and in the availability of acetyl group acceptor.

Meikle and colleagues argued that the *N*-acetyltransferase reaction occurs via a random-order mechanism (*Figure 5C*). They demonstrated that both *N*-acetyl-α-D-glucosamine and COA are required for the reverse reaction, suggesting that the formation of a ternary complex of acetyl-CoA, α-D-glucosamine, and the enzyme enables the reaction to proceed in a single step without the requirement of an acetylated-enzyme intermediate. They also report two $K_m$ values, that differ by 5–10 times, for both acetyl-CoA and glucosamine substrates, suggesting that both protomers, perhaps, bind to the substrates at varying affinity and only one of the protomers is catalyzing the acetyl transfer at any given point of time (*Meikle et al., 1995*). Mahuran and colleagues showed that acetylated enzyme intermediate could not be identified by affinity pulldown of purified transmembrane *N*-acetyltransferase after incubating it with [$^3$H] acetyl-CoA and proposed that the formation of a stable acetylated enzyme intermediate is not required for the reaction to proceed (*Fan et al., 2011*). Our structure does not provide evidence of H269 acetylation (*Figure 3*). However, we observe that acetyl-CoA binds tightly to HGSNAT endogenously resulting in a stable HGSNAT-acetyl-CoA intermediate that stayed as an intact complex all through the affinity purification, size-exclusion chromatography, and cryo-EM sample preparation steps even without the addition of exogenous acetyl-CoA to any of the buffers (*Figure 5—figure supplement 1*). Perhaps, this tight endogenous binding of HGSNAT is the reason why Mahuran and colleagues could not observe the labeling of purified transmembrane *N*-acetyltransferase by [$^3$H] acetyl-CoA. The structure that we obtained does not by itself settle the debate about the mechanism of HGSNAT-mediated acetylation. Our structure is in a conformation that could proceed through any of the three enzyme-catalyzed bisubtrate reaction mechanisms for acetyl transfer (*Figure 5*). While we were revising our manuscript, another group published structures of apo HGSNAT and HGSNAT-ACO, HGSNAT-COA-NAG complexes. All these structures were obtained at pH 8.0 in 1% GDN and revealed no drastic conformational isomerization in the protein except for localized motion in TM2 and TM3 and residues at ACOS (*Xu et al., 2024*). Xu and colleagues do not show if the order of binding of the substrate is crucial for catalysis, and nor do they disprove the existence of acetylated enzyme intermediate. We believe structures of HGSNAT at lysosomal luminal pH in the presence and absence of an acetyl group acceptor will demonstrate not only if the enzyme gets acetylated during the reaction but also show if there are pH-driven conformational changes within the protein. It is also likely that the choice of detergent used determines the conformational isomerization. So, obtaining structures in membrane mimetics like nanodiscs in the presence of native HGSNAT lipids could provide a clearer insight into HGSNAT catalyzed reaction mechanism.

HGSNAT is believed to be produced as a pro-protein that gets proteolytically processed into its mature form where two fragments of unequal sizes, the α-HGSNAT and β-HGSNAT, continue to stay together because of a disulfide bond mediated interaction between C123 (β6) in the N-terminal LD of the α-HGSNAT and C434 in the LL3 of the *C*-terminal TMD of β-HGSNAT (*Durand et al., 2010*).

However, according to the structure, these amino acids are ~34 Å apart making such a bond impossibility in the acetyl-CoA bound HGSNAT conformation reported here (*Figures 1A, D and 2B*). Moreover, we notice that C434 is disulfide bonded to C415 on LL3. C123 of β6 is predicted to form a disulfide bond with C151 of β8, and we do not see density to model such a disulfide. A bond between C123 of LD and C434 of LL3 could be a possibility only if there is a drastic conformational change during that brings LD closer to the LL3. Thus, we believe that the α-HGSNAT and β-HGSNAT stay together because of a series of hydrogen bonds, dipole-dipole and hydrophobic interactions between β2-β3 turn, β4-β5 turn, LL1, LL3, and LL4 (*Figure 2* and *Figure 2—figure supplement 2*).

The protease that cleaves HGSNAT into α-HGSNAT and β-HGSNAT is unknown, and the site for proteolysis is unclear. Fan et al, proposed that the site of proteolysis is between amino acids N144 and G145 in the LD (*Fan et al., 2011*). N144-G145 is located on the β7-β8 turn of the LD, a region poorly resolved in our structure (*Figures 1A and 2A*). Although proteolysis at this position will cleave the LD into two parts (β1–7 and β8), the LD should remain attached to TMD because of the extensive network of interactions along the LD-TMD interface (*Figure 2B* and *Figure 2—figure supplement 2*). Durand et al, proposed that the site of proteolysis is between the end of the luminal domain and the beginning of LL1 (*Durand et al., 2010*). Sequence-based protease site prediction in Procleave and Prosper servers, using the sequence between β8 of LD and LL1, suggests multiple potential sites of proteolysis, for example, L154-A155 at the C-term of β8, S162-N163 in the loop connecting LD to TM1, F170-L171 at the N-term of TM1, L187-S188, and W231-R232 at the N- and C-term of the CL1 (*Li et al., 2020*; *Song et al., 2012*). It is hard to speculate which of these are most probable, as the lysosomes are rich in aspartic, serine, and cysteine proteases, including intra-membrane proteases that are involved in protein degradation and proprotein processing. In addition, some matrix metalloproteinases have also been shown to act in intracellular compartments and have been implicated in MPS diseases (*Batzios et al., 2012*; *Jobin et al., 2017*; *Müller et al., 2012*; *Schröder and Saftig, 2016*; *Turk et al., 2000*). The recombinant HGSNAT that we produced is a non-proteolyzed form of HGSNAT, as we see a band corresponding to full-length HGSNAT in our SDS-PAGE analysis and a peak corresponding to an intact full-length dimer in our FSEC analysis. HGSNAT is encoded by 18 exons, where the first 6 exons encode an N-terminal LD & TM1, which are only present in the metazoans. The remaining 10 TMs and the C-terminus are well conserved in archaea, bacteria, and plants (*Fan et al., 2006*; *Hrebícek et al., 2006*). Based on the evolutionary sequence and structure conservation and conformational flexibility observed in CL1, we speculate that LD & TM1 together form α-HGSNAT and TMs 2–11 form β-HGSNAT (*Figures 2A and 4A*, and *Figure 2—figure supplement 1A*).

There are no known homologs of the LD, and even the strand arrangement in two sheets appears to be rare (*Koch et al., 1992*). A structure-based search in the Dali server lists hits with <15% identity, with immunoglobulin-like & transthyretin folds as top hits (*Supplementary file 1*; *Beale et al., 2015*; *Felisberto-Rodrigues et al., 2011*; *Parker et al., 2021*; *Rajasekar et al., 2019*). Unlike the LD, a typical immunoglobulin-like fold is a disulfide-containing β-sandwich made of 7–9 β-strands arranged in two anti-parallel β-sheets, with a conserved core formed by four strands β2, β3, β5, & β6 (*Figure 2—figure supplement 1C and D*). In the Ig-like fold, the β2 and β5 strands are in one sheet, and the β3 and β6 strands are in the second one, with β2 and β6 being linked by a disulfide bond. The remaining 3–5 strands comprise the remaining variable region of the Ig-like fold (*Bork et al., 1994*; *Chidyausiku et al., 2022*). Although the % identity of LD and the transthyretin fold containing hits is <15%, the strand composition of the two sheets that make the β-sandwich is identical. However, a typical transthyretin fold contains a conserved short α-helix between β5 & β6 and often does not contain disulfide bonds (*Figure 2—figure supplement 1C and D*; *Hörnberg et al., 2000*). Another β-sandwich that shows a similar secondary structure assignment as LD is the type-II C2 domain, where the top sheet is made of β1, β4, β7, & β8 strands while the bottom sheet is made of β2, β3, β5, & β6 (*Hirano et al., 2019*; *Kretsinger et al., 2013*). However, both sheets in the type-II C2 domain are anti-parallel, and the arrangement of the top strand is β4-β1-β8-β7. In LD, one sheet is anti-parallel, and the other is a mixed sheet where the arrangement in the top sheet is β4-β1-β7-β8 (*Figure 2—figure supplement 1C and D*). Because of these non-trivial dissimilarities of LD with three major types of β-sandwiches, we believe that LD is a new fold of β-sandwich that could be best described as a 'transthyretin-like' fold. In fact, two proteins in the top hits from a structure-based similarity search on the Dali server show the same sheet arrangement as LD (*Supplementary file 1* and *Figure 2—figure supplement 1D*). One of these proteins is AlgF (PDB ID: 6CZT), an adaptor protein from Pseudomonas believed to be involved

in O-acetylation of alginate exopolysaccharides and the other is an SPH (self-incompatibility protein homolog) domain (PDB ID: 6G7G) that is described as transthyretin-like and is believed to be a plant secreted protein involved in cell death (*Figure 2—figure supplement 1D*).

The function of LD remains unknown. DeepSite predicts the presence of two ligand binding sites in HGSNAT, one that overlaps with the ACOS and the other in the cavity between the two sheets of LD (*Figure 3—figure supplement 1B*; *Jiménez et al., 2017*). However, it is unclear if LD binds to a ligand and/or a metal ion. It has been shown that the *N*-acetyltransferase activity of HGSNAT is enhanced in the presence of anionic phospholipids (*Fan et al., 2011*). However, we expect lipids to bind towards the cytosolic side at the dimer interface. We see unexplained density in these regions that could be best explained as ordered lipids or digitonin (*Figure 2—figure supplement 3*). A recent report of a computational modeling and molecular dynamics simulation study conducted on a model of OafB, a bacterial O-antigen modifying transmembrane acetyltransferase of the ATAT family within the TmAT superfamily, showed that the periplasmic SGNH domain undergoes large conformational changes and aid in O-antigen acetylation (*Newman et al., 2023*). OafB has two domains – the transmembrane acetyltransferase-3 domain (AT-3) and the periplasmic SGNH domain. Although the domain organization is similar to HGSNAT, there are marked differences in the structure (*Figure 2—figure supplement 1B*). AT-3 domain and HGSNAT TMD are not homologous, even though they share a similar architecture to the ACOS. The LD of HGSNAT is different from the SGNH domain, in the sense that LD is a β-sandwich at the N-terminus of the protein while SGNH is a α-β-α sandwich with a single β-sheet sandwiched between two helical domains at the C-terminus of the protein. There are non-trivial differences between the predicted structures of ATAT and YeiB family members. However, it is possible that the TMDs and the extra-membranous domains function similarly. It is likely that LD of HGSNAT binds to HS, stabilizing the terminal non-reducing sugar of HS near the luminal side of ACOS for catalysis and remains to be structurally investigated. In fact, it has been shown that as the size of the acetyl group acceptor was increased in an acetyltransferase reaction from mono- to di- to tetra-saccharide, the $K_m$ values decreased from 0.6 mM to 7 µM (*Meikle et al., 1995*).

The high-resolution structure of HGSNAT reported in this study heralds a new beginning for structural exploration of the TmAT superfamily. The conformation of HGSNAT observed in our cryo-EM studies does not put to rest the debate on the kind of bi-substrate reaction mechanism that HGSNAT follows to catalyze the acetyl transfer. However, our structure underscores the requirement to characterize the remaining states of the enzymatic reaction. The molecular basis we delineated for the mutation-induced dysfunction in HGSNAT will serve as a blueprint for the structure-based design of novel therapeutic modulators that could rescue the function of milder variants.

# Materials and methods

## Key resources table

| Reagent type (species) or resource | Designation | Source or reference | Identifiers | Additional information |
|---|---|---|---|---|
| Gene (*Homo sapiens*) | Heparan-α-glucosaminide *N*-acetyltransferase | GenScript | NCBI Reference Sequence NM_152419.3 UniProt ID: Q68CP4-2 | Isoform-2 of HGSNAT |
| Cell line (*Homo sapiens*) | HEK293S GnTI- | ATCC | Cat # ATCC CRL-3022 | Used for protein expression |
| Cell line (*Spodoptera frugiperda*) | SF9 | Gibco | Cat # 12659017 | Used for baculovirus production |
| Strain, strain background (*Escherichia coli*) | DH10Bac | Thermo Fisher | 10361012 | Chemically competent cells for Bacmid production |
| Transfected construct (*Homo sapiens*) | N-Strep-tag-II-GFP-HGSNAT (Isoform-2) | GenScript (This study) | | Expression construct used for large-scale protein production |
| Recombinant DNA reagent | pEG BacMam (plasmid) | Addgene | Plasmid # 160683 | Expression vector for cloning HGSNAT with an N-terminal GFP tag |
| Chemical compound, drug | Desthiobiotin | Iba life sciences | Cat # 2-1000-002 | Modified biotin for Strep-Tactin elution buffer |
| Software, algorithm | CryoSPARC v4.2.1 | PMID:28165473 | RRID:SCR_016501 | https://cryosparc.com/ |
| Software, algorithm | ResMap | PMID:24213166 | | https://resmap.sourceforge.net/ |
| Software, algorithm | ModelAngelo | PMID:38408488 | | https://sbgrid.org/software/titles/modelangelo |
| Software, algorithm | Phenix | PMID:31588918 | RRID:SCR_014224 | https://phenix-online.org/ |

*Continued on next page*

*Continued*

| Reagent type (species) or resource | Designation | Source or reference | Identifiers | Additional information |
|---|---|---|---|---|
| Software, algorithm | Coot | PMID:20383002 | RRID:SCR_014222 | https://www2.mrc-lmb.cam.ac.uk/personal/pemsley/coot/ |
| Software, algorithm | ChimeraX | PMID:37774136 | RRID:SCR_015872 | https://www.cgl.ucsf.edu/chimerax/ |
| Software, algorithm | FoldX | PMID:15980494 | RRID:SCR_008522 | https://foldxsuite.crg.eu/ |
| Software, algorithm | ConSurf Database | PMID:27166375 | RRID:SCR_002320 | http://consurfdb.tau.ac.il/ |
| Software, algorithm | AlphaFold | PMID:34791371 | RRID:SCR_023662 | https://alphafold.ebi.ac.uk/ |
| Software, algorithm | DALI | PMID:36419248 | RRID:SCR_013433 | http://ekhidna2.biocenter.helsinki.fi/dali/ |
| Other | TurboFect | Thermo Fisher | Cat # R0534 | For transfecting HEK293S GnTI- |
| Other | Cellfectin II | Thermo Fisher | Cat # 10362100 | For transfecting Sf9 cells |
| Other | DMEM | Corning | Cat # 10–013-CV | For adherent cell culture |
| Other | Sf-900 III | Gibco | Cat # 12658027 | For Sf9 culture |
| Other | FreeStyle 293 | Gibco | Cat # 12338026 | For Hek293S GnTI- culture |
| Other | Superose 6 Increase 10/300 GL | Cytiva | Cat # GE29-0915-96 | Column for size-exclusion chromatography |
| Other | Chromolith RP-18 | Supelco | Cat # 102129 | Column for LC-MS |
| Other | Strep-Tactin Superflow high-capacity resin | Iba life sciences | Cat # 2-1208-025 | Resin for affinity purification |
| Other | CHAPS | Anatrace | Cat # C316 | Detergent |
| Other | n-octyl-β-D-glucoside | Anatrace | Cat # O311 | Detergent |
| Other | Lauryl maltose neopentyl glycol | Anatrace | Cat # NG310 | Detergent |
| Other | n-dodecyl-β-D-maltooside | Anatrace | Cat # D310S | Detergent |
| Other | Glyco-diosgenin | Anatrace | Cat # GDN101 | Detergent |
| Other | Digitonin | Sigma | Cat # 300410 | Detergent |
| Other | UltrAufoil R 1.2/1.3, 300 mesh, Au holey-gold grids | Electron Microscopy Sciences | Cat # Q350AR13A | Cryo-EM grids |

## Cloning and site-directed mutagenesis

The codon-optimized gene encoding the isoform 2 of full-length human HGSNAT was synthesized by GenScript. The synthesized gene was then cloned into the pEG BacMam expression vector (Addgene plasmid # 160683) between EcoRI and NotI restriction sites, to be expressed via baculoviral transduction in HEK293S GnTI- cells (ATCC # CRL-3022) as a fusion protein containing an N-terminal Strep-tag-II-GFP. The integrity of the clone was confirmed by Sanger sequencing (Plasmidsaurus, OR). HGSNAT variants discussed in the manuscript were prepared in the pEG BacMam expression vector background by GenScript.

## Cell culturing, transient transfection, and transduction

Adherent HEK293S GnTI- cells were grown in Dulbecco's Modified Eagle Medium (DMEM, Croning) supplemented with 10% fetal bovine serum (FBS, Gibco) at 37 °C. Immediately preceding transfection, the cells were washed with 1 X PBS (Gibco) and supplied with fresh pre-warmed DMEM containing 10% FBS. $1 \times 10^6$ cells were transfected with 1 µg DNA using TurboFect (Thermo Fisher Scientific) suspended in serum-free DMEM as suggested by the manufacturer's protocol. The transfected cells were grown at 37 °C and 5% $CO_2$ for 8–10 hr. The cell-culture media was then replaced with fresh pre-warmed DMEM containing 10% FBS and 10 mM of sodium butyrate, and the cells were grown at 32 °C and 5% $CO_2$ for an additional 24–36 hr, before harvesting. All transfections were done at 80% confluency. Transfected cells were used to screen for ideal expression and purification conditions.

Baculovirus preparation was done as described by Goehring and colleagues (*Goehring et al., 2014*). Briefly, DH10Bac cells (Thermo Fisher Scientific) were transformed with an HGSNAT expression vector, and lacZ- colonies were selected on gentamycin-kanamycin-tetracycline LB agar plates for bacmid DNA isolation. $1 \times 10^6$ adherent Sf9 cells (Gibco # 12659017) grown in serum-free Sf-900 III media (Gibco) were transfected with 1 µg of bacmid using Cellfectin II reagent (Gibco) per the manufacturer's protocol. Transfected cells were grown at 27 °C for 96 hr. The supernatant media from

the cells was harvested, filtered through a 0.2 µm filter, and stored as P1 virus. 100 µL of P1 virus was added to 1 L of Sf9 cells at a cell density of $1\times10^6$/mL in serum-free Sf-900 III media. The cells were grown at 96 hr at 27 °C while shaking at 120 rpm. The cells were spun down at 4000×g for 20 min, and the supernatant media was filtered through a 0.2 µm filter and stored as P2 baculovirus. P2 baculovirus was used for large-scale transduction of HEK293S GnTI⁻ cells. Large-scale expression of HGSNAT was done by baculoviral transduction of HEK293S GnTI⁻ cells suspended in FreeStyle 293 expression media containing 2% FBS. Mammalian cell culture at a density of $3\times10^6$ cells/mL was transduced using P2 baculovirus at a multiplicity of infection of 1.5–2 and was incubated on an orbital shaker at 37 °C and 5% $CO_2$ for 8–10 hr. The cells were supplemented with 10 mM sodium butyrate and were then incubated in a shaker at 32 °C and 5% $CO_2$ for an additional 38–40 hr. The cells were harvested by centrifugation at 4000×g for 10 min, and the cell pellet was stored at –80 °C until further use.

## Protein expression and thermostability analysis

To identify suitable conditions for large-scale expression and solubilization of HGSNAT, we employed fluorescence-detection size-exclusion chromatography (FSEC) (*Kawate and Gouaux, 2006*). Briefly, 100,000 transfected cells were solubilized in 500 µl of 1% detergent, 25 mM Tris-HCl, pH 7.5, 200 mM NaCl, 1 mM PMSF, 0.8 µM aprotinin, 2 µg/mL leupeptin, and 2 µM pepstatin A at 4 °C for 1 hr on an end-end rotator. After solubilization, the lysate was centrifuged at 185,000×g for 1 hr at 4 °C. The supernatant was filtered through 0.45 µm filter, and 100 µl filtrate was analyzed on a Superose 6 Increase 10/300 GL column (Cytiva Life Sciences) pre-equilibrated with 0.15 mM LMNG, 25 mM Tris-HCl, pH 7.5, and 200 mM NaCl. GFP fluorescence in the eluate was monitored by a fluorometer (Shimadzu scientific instruments) set at Ex/Em of 485/510 nm. To test the stability of HGSNAT in various conditions, the solubilized samples were heated at 55 °C for 15 min, centrifuged at 10,000 rpm for 10 min, filtered through 0.45 µm filter, and were analyzed again on a Superose 6 Increase 10/300 GL column. To measure the overall relative expression of mutants, 100,000 cells expressing the mutants were solubilized in 100 ul of 1% digitonin, 25 mM Tris-HCl, pH 7.5, 200 mM NaCl, 1 mM PMSF, 0.8 µM aprotinin, 2 µg/mL leupeptin, and 2 µM pepstatin A at 4 °C for 1 hr on an end-end rotator. The solubilized lysates were transferred to Costar 96-well flat bottom clear plates without centrifugation, and GFP fluorescence was monitored at Ex/Em of 480/520 nm in SpectroMax M5 (Molecular Devices) microplate reader. For FSEC analysis, mutants were solubilized in 1% digitonin, but were also centrifuged and filtered before being analyzed on Superose 6 Increase 10/300 GL column. Thermal stability of the mutants was tested by heating solubilized lysates of the mutants and WT HGSNAT at 65 °C for 15 min and analyzing them by FSEC.

## Purification of HGSNAT

Cell debris from the sonicated lysate of HEK293S GnTI⁻ cells expressing N-terminal GFP fusion of HGSNAT was removed by centrifugation at 2400×g for 10 min. The supernatant from the low-speed centrifugation step was subjected to ultra-centrifugation at 185,000×g for 1 hr at 4 °C to harvest membranes. Membranes were resuspended using a dounce homogenizer in suspension buffer (25 mM Tris-HCl, pH 7.5, 200 mM NaCl, 1 mM PMSF, 0.8 µM aprotinin, 2 µg/mL leupeptin, and 2 µM pepstatin A). To this membrane suspension, an equal volume of 2% digitonin solution prepared in membrane suspension buffer was added. The membrane suspension was solubilized for 90 min. at 4 °C. The solubilized membrane suspension was centrifuged at 185,000×g for 1 hr at 4 °C. The solubilized supernatant was passed through the Strep-Tactin affinity resin (IBA Life Sciences) column at a flow rate of ~0.3–0.5 ml/min. Affinity resin saturated with HGSNAT was washed with 5–6 column volumes of 0.5% digitonin, 25 mM Tris-HCl, pH 7.5, and 200 mM NaCl. The bound protein was eluted using 5 mM D-desthiobiotin prepared in the wash buffer. The purity and homogeneity of purified HGSNAT was confirmed by performing FSEC and SDS-PAGE. The amino acid sequence of purified HGSNAT was verified by peptide mass fingerprinting of the bands on SDS-PAGE. For FSEC experiments, ~10 µL of elution fractions were loaded onto a Superose 6 Increase 10/300 GL column pre-equilibrated with 0.15 mM LMNG, 25 mM Tris-HCl, pH 7.5, and 200 mM NaCl. The elution fractions containing homogeneous and pure protein were pooled and concentrated to 2 mg/mL. The concentrated protein was further purified by size-exclusion chromatography (SEC) using Superose 6 Increase 10/300 GL column pre-equilibrated with 0.5% digitonin, 25 mM Tris-HCl, pH 7.5, and 200 mM NaCl. HGSNAT

corresponding to the dimeric peak from the SEC step was pooled and concentrated to 0.9 mg/mL for cryo-EM sample preparation.

## LC-MS analysis of purified HGSNAT

To identify the endogenously bound acetyl-CoA in recombinant HGSNAT, the purified protein was loaded onto a Chromolith RP-18 end-capped column (4.6 mm X 100 mm; Supleco) operated at 1 ml/min flow rate and 40 °C. Acetyl-CoA was eluted by a gradient of mobile phase A (5 mM ammonium formate) and mobile phase B (10–90% acetonitrile). The eluted samples were analyzed by a Thermo TSQ-Quantis (Thermo Scientific) in positive mode at 5.5kV capillary voltage while scanning at 1000 Da/s from 200 to 1000 m/z in single reaction monitoring (SRM) mode. The data was analyzed using Thermo FreeStyle 1.6 (Thermo Scientific), using an SRM filtering for acetyl-CoA precursor (810.1 m/z) and product (303.1 m/z). To monitor the specific endogenous binding of acetyl-CoA to HGSNAT, we also purified apo HGSNAT from the membrane suspension dialyzed in membrane suspension buffer for 4 days at 4 °C. Apo HGSNAT was processed for LC-MS analysis in the same fashion as the HGSNAT-ACO complex.

## Cryo-EM sample preparation and data collection

UltrAuFoil holey-gold 300 mesh 1.2/1.3 μm size/hole space grids (UltrAuFoil, Quantifoil) were glow discharged, using a PELCO glow discharger, for 1 min at 15 mA current and 0.26 mBar air pressure and were immediately used for sample vitrification. 2.5 μL of HGSNAT at 0.9 mg/ml was applied to the glow-discharged grids, and subsequently blotted for 2 s at 18 °C and 100% humidity using a Vitrobot (mark IV, Thermo Fisher Scientific). Without any wait time, the grids were plunge-frozen in liquid ethane. Movies were recorded on a Gatan K3 direct electron detector in super-resolution counting mode with a binned pixel size of 0.85 Å per pixel using Serial EM on a FEI Titan Krios G4i transmission electron microscope operating at an electron beam energy of 300 KeV with a Gatan Image Filter slit width set to 20 eV. Exposures of ~2 s were dose fractionated into 50 frames, attaining a total dose of ~50 $e^-$ Å$^{-2}$, and the defocus values were varied from –1.0 to –2.5 μm.

## Cryo-EM data processing

All image processing was done using CryoSPARC (v 4.2.1) unless specified otherwise (*Punjani et al., 2017*). Briefly, motion correction was carried out on the raw movies, and subsequently, the contrast transfer function (CTF) was estimated using patch-CTF correction on dose-weighted motion-corrected averages within cryoSPARC (*Figure 1—figure supplement 2*). A total of 15,067 micrographs were collected, of which 10,394 were selected for further processing based on the CTF fit. Both blob picking and template picking were used to pick particles, and duplicates were removed. Particles were extracted initially using a box size of 288 pixels. A subset of the particles picked by the blob picker was used to generate a low-resolution *ab initio* reconstruction, which was later used as a reference for iterative 3D classification and heterogenous refinement. The pooled particles were subjected to multiple rounds of reference-free 2D classification and heterogenous refinement to get a cleaned subset of particles which resulted in classes with recognizable features. The cleaned particles were re-extracted with a box size of 360 pixels, and a stack of 94,875 good particles was selected and subjected to an initial round of NU refinement using an *ab initio* model as a reference (*Punjani et al., 2020*). 86,500 particles resulted in a 3.89 Å map, which had well-resolved luminal domain architecture, and visible secondary structure features of TM helices. To improve the resolution further, only the particles belonging to micrographs with CTF fit <4 Å were selected for subsequent refinements. Both C1 and C2 symmetry-imposed maps overlapped well, and we used C2 symmetry in the final rounds of processing (*Figure 1—figure supplements 2 and 3*). A final subset of 57,739 particles was used for NU-refinement, yielding a 3.26 Å map (Fourier shell coefficient (FSC)=0.143 criterion). Symmetry expansion and local refinement with a focused mask on the LD and TMD domains was performed to improve local density in these domains. The map obtained by symmetry expansion was only used to assess the fit of the model to density. A composite map generated by combining these local maps was used to finalize the fit of the side chains during model building and refinement (*Figure 1—figure supplement 3D*). Local resolutions were estimated using the RESMAP software (*Kucukelbir et al., 2014*).

## Model building, refinement, and structure analysis

ModelAngelo was used to build HGSNAT isoform 2 into the final map, and final refinement was performed in Phenix (*Jamali et al., 2023*; *Liebschner et al., 2019*). Coot was used to rebuild certain portions of the protein and to analyze the Ramachandran outliers (*Emsley et al., 2010*). ChimeraX was used for making the figures (*Meng et al., 2023*). Ligand binding site prediction on HGSNAT was performed using MOLEonline and DeepSite (*Jiménez et al., 2017*; *Pravda et al., 2018*). To analyze the effect of missense mutations on HGSNAT and to calculate free energies indicating the relative mutation-induced destabilization, we used FoldX (*Schymkowitz et al., 2005*).

## Acknowledgements

We thank the University of Michigan (UM) cryo-EM facility staff for assistance with cryo-EM data collection, and Drs. Michael Cianfrocco, Farzad Jalali-Yazdi, Jonathan Coleman, and Prashant Rao for helpful comments on cryo-EM analysis. UM, cryo-EM facility receives generous support from UM BSI and UM LSI. We thank Dr. Carmen Dunbar and the Biological Mass Spectrometry Core at UM for the processing of LC-MS samples. We thank all the members of Mosalaganti lab and Baldridge lab for their educative discussions on our work. This work was supported by the start-up funding provided to SM by the University of Michigan. SM is a recipient of the NIH Director's New Innovator Award (DP2) (1DP2GM150019-01).

## Additional information

### Funding

| Funder | Grant reference number | Author |
|---|---|---|
| University of Michigan | Startup funding | Shyamal Mosalaganti |
| National Institutes of Health | 1DP2GM150019-01 | Shyamal Mosalaganti |

The funders had no role in study design, data collection and interpretation, or the decision to submit the work for publication.

### Author contributions

Vikas Navratna, Conceptualization, Data curation, Formal analysis, Supervision, Validation, Investigation, Visualization, Methodology, Writing - original draft, Project administration, Writing – review and editing; Arvind Kumar, Validation, Visualization, Writing – review and editing; Jaimin K Rana, Shyamal Mosalaganti, Data curation, Formal analysis, Supervision, Funding acquisition, Validation, Visualization, Project administration, Writing – review and editing

### Author ORCIDs

Vikas Navratna ⓘ https://orcid.org/0000-0001-8599-1461
Arvind Kumar ⓘ https://orcid.org/0000-0002-8421-8669
Shyamal Mosalaganti ⓘ https://orcid.org/0000-0002-5934-687X

Reviewer #1 (Public review): https://doi.org/10.7554/eLife.93510.3.sa1
Reviewer #2 (Public review): https://doi.org/10.7554/eLife.93510.3.sa2
Reviewer #3 (Public review): https://doi.org/10.7554/eLife.93510.3.sa3
Author response https://doi.org/10.7554/eLife.93510.3.sa4

## Additional files

### Supplementary files

• Supplementary file 1. Homologs of transmembrane domain (TMD) and luminal domain (LD) of heparan-α-glucosaminide N-acetyltransferase (HGSNAT) found in a Dali search, Dali (**D**istance **Ma**trix **Ali**gnment) web server was used to search the existing database of known structures to

find homologs of HGSNAT (*Holm et al., 2023*). The poor % identity (<20%) and low % sequence alignment suggests that there are no available structures of homologs of HGSNAT. Low mean RMSD of hits obtained using LD of HGSNAT as input suggests that LD is like some of the existing β-sandwiches, but TMD of HGSNAT is a novel fold.

• Supplementary file 2. List of heparan-α-glucosaminide N-acetyltransferase (HGSNAT) mutations implicated in mucopolysaccharidosis IIIC (MPS IIIC). FoldX web server was used to predict relative mutant stability (*Schymkowitz et al., 2005*). Positive total energy value indicates destabilization, with greater values meaning lower stability. Nonsense mutations indicated in the *Figure 4* as black were not included in FoldX calculations. Polymorphisms are italicized. All other mutants listed are missense mutations (*Canals et al., 2011*; *Fan et al., 2006*; *Fedele and Hopwood, 2010*; *Feldhammer et al., 2009a*; *Feldhammer et al., 2009b*; *Hrebícek et al., 2006*; *Huizing and Gahl, 2020*).

• MDAR checklist

## Data availability

Structure coordinates and cryo-EM maps have been deposited and released under the accession codes RCSB PDB-8TU9; EMDB EMD-41620. Uncropped gel showing purification has been included as Figure1-Figure_supplement_1M-Source_Data 1 and Figure1-Figure_supplement_1M-Source_Data 2.

The following datasets were generated:

| Author(s) | Year | Dataset title | Dataset URL | Database and Identifier |
| --- | --- | --- | --- | --- |
| Navratna V, Kumar A, Mosalaganti S | 2023 | Cryo-EM structure of HGSNAT-acetyl-CoA complex at pH 7.5 | https://www.rcsb.org/structure/8TU9 | RCSB Protein Data Bank, PDB-8TU9 |
| Navratna V, Kumar A, Mosalaganti S | 2023 | Cryo-EM structure of HGSNAT-acetyl-CoA complex at pH 7.5 | https://www.ebi.ac.uk/emdb/EMD-41620 | ArryaExpress, EMD-41620 |

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
