## [Editor Report · eLife assessment]

This **important** study presents the structure of human heparan-alpha-glucosaminide N-acetyltransferase (HGSNAT) in the acetyl-CoA bound state, providing the first description of the architecture of this family of integral membrane enzymes, and revealing the mode of acetyl-CoA binding. The structural work is **convincing**, with a high resolution and isotropic single-particle cryoEM map and an atomic model that is well-justified by the density map, with strong density for the acetyl-CoA ligand. However, experimental support for the molecular mechanism of the HS acetylation reaction and the impact of disease-causing mutations is **incomplete**. This work will be of interest to biochemists and structural biologists studying the structure and function of integral membrane enzymes, as well as those interested in genetic diseases resulting from mutations in this family of enzymes, such as mucopolysaccharidosis IIIC (MPS III-C).

---

## [Referee Report · Reviewer #1 (Public review)]

This article by Navratna et al. reports the first structure of human HGSNAT in an acetyl-CoA-bound state. Through careful structural analysis, the authors propose potential reasons why certain human mutations lead to lysosomal storage disorders and outline a catalytic mechanism. The structural data are of good quality, and the manuscript is clearly written. This study represents an important step toward understanding the mechanism of HGSNAT and is valuable to the field. I have the following suggestions:

(1) The authors should characterize whether the purified protein is active. Otherwise, how does one know if the detergent used maintains the protein in a biologically relevant state? The authors should at least attempt to do so. If these prove to be challenging, at the very least, the authors should try a cell-based assay to demonstrate that the GFP tag does not interfere with the function.

(2) In Figure 5, the authors present a detailed schematic of the catalytic cycle, which I find to be too speculative. There is no evidence to suggest that this enzyme undergoes isomerization, similar to a transporter, between open-to-lumen and open-to-cytosol states. Could it not simply involve some movements of side chains to complete the acetyl transfer?

---

## [Referee Report · Reviewer #2 (Public review)]

Summary:

This work describes the structure of Heparan-alpha-glucosaminide N-acetyltransferase (HGSNAT), a lysosomal membrane protein that catalyzes the acetylation reaction of the terminal alpha-D-glucosamine group required for degradation of heparan sulfate (HS). HS degradation takes place during the degradation of the extracellular matrix, a process required for restructuring tissue architecture, regulation of cellular function and differentiation. During this process, HS is degraded into monosaccharides and free sulfate in lysosomes.

HGSNAT catalyzes the transfer of the acetyl group from acetyl-CoA to the terminal non-reducing amino group of alpha-D-glucosamine. The molecular mechanism by which this process occur has not been described so far. One of the main reasons to study the mechanism of HGSNAT is that multiple mutations spanning the entire sequence of the protein, such as, nonsense mutations, splice-site variants, and missense mutations lead to dysfunction that causes abnormal accumulation of HS within the lysosomes. This accumulation is a cause of mucopolysaccharidosis IIIC (MPS IIIC), an autosomal recessive neurodegenerative lysosomal storage disorder, for which there are no approved drugs or treatment strategies.

This paper provides a 3.26A structure of HGSNAT, determined by single-particle cryo-EM. The structure reveals that HGSNAT is a dimer in detergent micelles, and a density assigned to acetyl-CoA. The authors speculate about the molecular mechanism of the acetylation reaction, map the mutations known to cause MPS IIIC on the structure and speculate about the nature of the HGSNAT disfunction caused by such mutations.

Strengths:

The paper describes a structure of HGSNAT a member of the transmembrane acyl transferase (TmAT) superfamily. The high-resolution of a HGSNAT bound to acetyl-CoA is important for our understanding of HGSNAT mechanism. The density map is of high-quality, except for the luminal domain. The location of the acetyl-CoA allows speculation about the mechanistic role of multiple residues surrounding this molecule. The authors thoroughly describe the architecture of HGSNAT and map the mutations leading to MPS IIIC.

---

## [Referee Report · Reviewer #3 (Public review)]

Summary:

Navratna et al. have solved the first structure of a transmembrane N-acetyltransferase (TNAT), resolving the architecture of human heparan-alpha-glucosaminide N-acetyltransferase (HGSNAT) in the acetyl-CoA bound state using single particle cryo-electron microscopy (cryoEM). They show that the protein is a dimer, and define the architecture of the alpha- and beta-GSNAT fragments, as well as convincingly characterizing the binding site of acetyl-CoA.

Strengths:

This is the first structure of any member of the transmembrane acyl transferase superfamily, and as such it provides important insights into the architecture and acetyl-CoA binding site of this class of enzymes.

The structural data is of a high quality, with an isotropic cryoEM density map at 3.3Å facilitating building of a high-confidence atomic model. Importantly, the density for the acetyl-CoA ligand is particularly well-defined, as are the contacting residues within the transmembrane domain.

The structure of HSGNAT presented here will undoubtedly lay the groundwork for future structural and functional characterization of the reaction cycle of this class of enzymes.

Weaknesses:

While the structural data for the state presented in this work is very convincing, and clearly defines the binding site of acetyl-CoA, to get a complete picture of the enzymatic mechanism of this family, additional structures of other states will be required.

A weakness of the study is the lack of functional validation. The enzymatic activity of the enzyme characterized was not measured, and the enzyme lacks native proteolytic processing, so it is a little unclear whether the structure represents an active enzyme.

---

## [Author Response]

The following is the authors’ response to the current reviews.

**Public Reviews:**

**Reviewer #1 (Public Review):**
This article by Navratna et al. reports the first structure of human HGSNAT in an acetyl-CoA-bound state. Through careful structural analysis, the authors propose potential reasons why certain human mutations lead to lysosomal storage disorders and outline a catalytic mechanism. The structural data are of good quality, and the manuscript is clearly written. This study represents an important step toward understanding the mechanism of HGSNAT and is valuable to the field. I have the following suggestions:(1) The authors should characterize whether the purified protein is active. Otherwise, how does one know if the detergent used maintains the protein in a biologically relevant state? The authors should at least attempt to do so. If these prove to be challenging, at the very least, the authors should try a cell-based assay to demonstrate that the GFP tag does not interfere with the function.

We have addressed these concerns in the revised version and mentioned these efforts in our previous response letter. We’re briefly mentioning them here again. We attempted measuring HGSNAT catalyzed reaction by monitoring the decrease in acetyl-CoA in the presence of D-glucosamine (acetyl group acceptor) using a coupled enzyme acetyl-CoA assay kit from SIGMA (MAK039) that converts acetyl-CoA to a fluorescent product measurable at Ex/Em of 535/587 nm. We noticed a decrease in the level of acetyl-CoA (gray) upon the addition of HGSNAT (red) (Rebuttal figure 1).

**Author response image 1. sa4fig1:** Acetyl-CoA levels in absence and presence of HGSNAT purified in digitonin. Decrease in the levels of 10 M acetyl-CoA was measured in presence of 10 M D-glucosamine and 30 nM HGSNAT at pH 7.5.

While optimizing the assay, Xu et al. (2024, Nat Struct Mol Biol) published structural and biochemical characterization of HGSNAT, showing that detergent-purified HGSNAT is active. In addition, we have shown by cryo-EM that GFP-tagged HGSNAT that we purified in detergent was already bound to the endogenous substrate ACO, an observation that has been observed by Xu et al., as well. Finally, we performed LC-MS on GFP-tagged HGSNAT purified in detergent to detect bound ACO, which could be further removed by dialysis. These results have been included in Figure S9. The endogenous binding of ACO to HGSNAT in detergent suggests that neither the tag nor detergent are detrimental to the function.

(2) In Figure 5, the authors present a detailed schematic of the catalytic cycle, which I find to be too speculative. There is no evidence to suggest that this enzyme undergoes isomerization, similar to a transporter, between open-to-lumen and open-to-cytosol states. Could it not simply involve some movements of side chains to complete the acetyl transfer?

We have already changed this figure in our latest submission. Perhaps the changes made were not obvious while reviewing. We agreed with this reviewer that the enzyme could likely achieve catalysis by simple side chain movements without undergoing extensive isomerization steps, as depicted in Figure 5. In the absence of data supporting large movements during the acetyl transfer reaction, old Figure 5 appeared speculative. Hence, we have edited Figure 5 in the revised version of the manuscript based on the observations we made in this study, and different states shown in the figure do not show any conformational changes and only depict acetyl transfer.

**Reviewer #2 (Public Review):**
Summary:This work describes the structure of Heparan-alpha-glucosaminide N-acetyltransferase (HGSNAT), a lysosomal membrane protein that catalyzes the acetylation reaction of the terminal alpha-D-glucosamine group required for degradation of heparan sulfate (HS). HS degradation takes place during the degradation of the extracellular matrix, a process required for restructuring tissue architecture, regulation of cellular function and differentiation. During this process, HS is degraded into monosaccharides and free sulfate in lysosomes.HGSNAT catalyzes the transfer of the acetyl group from acetyl-CoA to the terminal non-reducing amino group of alpha-D-glucosamine. The molecular mechanism by which this process occur has not been described so far. One of the main reasons to study the mechanism of HGSNAT is that multiple mutations spanning the entire sequence of the protein, such as, nonsense mutations, splice-site variants, and missense mutations lead to dysfunction that causes abnormal accumulation of HS within the lysosomes. This accumulation is a cause of mucopolysaccharidosis IIIC (MPS IIIC), an autosomal recessive neurodegenerative lysosomal storage disorder, for which there are no approved drugs or treatment strategies.This paper provides a 3.26A structure of HGSNAT, determined by single-particle cryo-EM. The structure reveals that HGSNAT is a dimer in detergent micelles, and a density assigned to acetyl-CoA. The authors speculate about the molecular mechanism of the acetylation reaction, map the mutations known to cause MPS IIIC on the structure and speculate about the nature of the HGSNAT disfunction caused by such mutations.Strengths:The paper describes a structure of HGSNAT a member of the transmembrane acyl transferase (TmAT) superfamily. The high-resolution of a HGSNAT bound to acetyl-CoA is important for our understanding of HGSNAT mechanism. The density map is of high-quality, except for the luminal domain. The location of the acetyl-CoA allows speculation about the mechanistic role of multiple residues surrounding this molecule. The authors thoroughly describe the architecture of HGSNAT and map the mutations leading to MPS IIIC.
**Reviewer #3 (Public Review):**
Summary:Navratna et al. have solved the first structure of a transmembrane N-acetyltransferase (TNAT), resolving the architecture of human heparan-alpha-glucosaminide N-acetyltransferase (HGSNAT) in the acetyl-CoA bound state using single particle cryo-electron microscopy (cryoEM). They show that the protein is a dimer, and define the architecture of the alpha- and beta-GSNAT fragments, as well as convincingly characterizing the binding site of acetyl-CoA.Strengths:This is the first structure of any member of the transmembrane acyl transferase superfamily, and as such it provides important insights into the architecture and acetyl-CoA binding site of this class of enzymes.The structural data is of a high quality, with an isotropic cryoEM density map at 3.3Å facilitating building of a high-confidence atomic model. Importantly, the density for the acetyl-CoA ligand is particularly well-defined, as are the contacting residues within the transmembrane domain.The structure of HSGNAT presented here will undoubtedly lay the groundwork for future structural and functional characterization of the reaction cycle of this class of enzymes.Weaknesses:While the structural data for the state presented in this work is very convincing, and clearly defines the binding site of acetyl-CoA, to get a complete picture of the enzymatic mechanism of this family, additional structures of other states will be required.A weakness of the study is the lack of functional validation. The enzymatic activity of the enzyme characterized was not measured, and the enzyme lacks native proteolytic processing, so it is a little unclear whether the structure represents an active enzyme.
**Recommendations for the authors:**

**Reviewer #3 (Recommendations For The Authors):**
In the response to reviewers, the authors mention revised coordinates, but the revised coordinates provided to this reviewer do not reflect the stated changes (I assume a technical error somewhere)

Perhaps, the old coordinates in the deposition system were resubmitted with the revised draft. Nevertheless, we have made the changes suggested by this reviewer to structure in the previous round and have released the new coordinates (PDB ID: 8TU9).

Is there any evidence for the interprotomer disulfide except for the map? e.g. if it is a disulfide-linked dimer, one should see a shift in mobility on non-reducing vs reducing SDS-PAGE. Without this, the evidence from the map is not conclusive - while the symmetry-related cysteines are nearby to one another, based on the map I could argue that they could just as well be modeled with the cys sidechains reduced and pointing away from one another.

In addition to building the density based on cryo-EM maps, we have performed FSEC-based thermal melt analysis of the Ala mutation of C334 that is involved in disulfide at the dimer interface. C334A is still expressed as a dimer, suggesting that C334A is not the only residue stabilizing the dimer. Upon heating the detergent-solubilized protein, we noticed that the FSEC peak for C334A shows a monomeric HGSNAT (Figure 4-Figure supplement 1 in main manuscript). We hypothesize that in the absence of C334 disulfide, the extensive hydrophobic side-chain interaction network displayed in Figure 2C is responsible for maintaining the integrity of the dimer. Heating disturbs these non-disulfide interactions, thereby rendering the protein monomer. We have also performed PAGE analysis as suggested by this reviewer and noticed that reducing conditions result in a monomeric protein band (Rebuttal figure 2). While we were revising this manuscript, two other groups published structures of HGSNAT (Xu et al., 2024, Nat. Struct Mol Biol, and Zhao et al., 2024, Nat. Comm). These groups have also identified this disulfide at the dimer interface in their HGSNAT structures. Zhao et al. showed that this disulfide is not crucial for dimerization and also suggested that it can break depending on the conformation of HGSNAT. Our FSEC results agree with this observation.

**Author response image 2. sa4fig2:** Comparison of purified HGSNAT on native and reducing SDS-PAGE. The arrows on both the gels indicate N-GFP-HGSNAT. The two bands on the SDS PAGE are, perhaps, two differentially glycosylated forms of HGSNAT.

The following is the authors’ response to the original reviews.

(1) The authors should characterize whether the purified protein is active. Otherwise, how does one know if the detergent used maintains the protein in a biologically relevant state? The authors should at least attempt to do so. If these prove to be challenging, at the very least, the authors should try a cell-based assay to demonstrate that the GFP tag does not interfere with the function. The authors would need to establish an in vitro assay using purified protein and assess the level of Acetyl-CoA in the reaction (there are commercial kits and a long list of literature showing how to measure this). They could also follow the HS acetylation reaction by e.g. HPLC-MS or NMR (among other methods).

The cryo-EM sample was prepared without the exogenous addition of ligand, as noted in the manuscript. However, we see that acetyl-CoA was intrinsically bound to the protein, indicating the ability of GFP-tagged HGSNAT protein to bind the ligand. Upon dialysis, we see release of acetyl-CoA from the protein, which we have confirmed by LC-MS analysis (Fig S9). We purified the protein at a pH optimal for acetyl-CoA binding, as suggested by Bame, K. J. and Rome, L. H. (1985) and Meikle, P. J. et al., (1995). Because we see acetyl-CoA in a structure obtained using a GFP fusion, we argue that GFP does not interfere with protein stability and ability to bind to the co-substrate. As demonstrated by existing literature HGSNAT catalyzed reaction is compartmentalized spatially and conditionally. The binding of acetyl-CoA happens towards the cytosol and is optimal at pH 7-0.8.0, while the transfer of the acetyl group to heparan sulfate occurs towards the luminal side and is optimal at pH 5.0-6.0. We attempted measuring HGSNAT catalyzed reaction by monitoring decrease in acetyl-CoA in presence of D-glucosamine (acetyl group acceptor) using a coupled enzyme acetyl-CoA assay kit from SIGMA (MAK039) that converts acetyl-CoA to a fluorescent product measurable at Ex/Em of 535/587 nm. We noticed a decrease in the level of acetyl-CoA in the presence of HGSNAT-ACO complex (blue) and apo HGSNAT (red); the difference compared to the ACO standard (gray) was not significant. While optimizing the assay, Xu et al. (2024, Nat Struct Mol Biol) published structural and biochemical characterization of HGSNAT, showing that detergent-purified HGSNAT is active.

**Author response image 3. sa4fig3:** Acetyl-CoA levels in absence and presence of HGSNAT purified in digitonin. Decrease in the levels of 10 mM acetyl-CoA was measured in presence of 10 mM D-glucosamine and 30 nM HGSNAT at pH 7.5.

(2) In Figure 5, the authors present a detailed schematic of the catalytic cycle, which I find to be too speculative. There is no evidence to suggest that this enzyme undergoes isomerization, similar to a transporter, between open-to-lumen and open-to-cytosol states. Could it not simply involve some movements of side chains to complete the acetyl transfer? The speculative nature of this assumption needs to be clearly acknowledged throughout the manuscript and discussed in more detail. The authors could use HDX-MS or introduce cysteine residues in the hypothetical inward- and outward-facing cavities and test accessibility by incubating the purified protein with maleimides or other agents reacting with free cysteine.

We thank the reviewers for this insightful critique. Yes, the enzyme could likely achieve catalysis by simple side chain movements without undergoing extensive isomerization steps, as depicted in Figure 5. We also agree with the reviewer that HDX-MS could be the best way to monitor the substrate-induced conformational dynamics within HGSNAT experimentally. In the absence of data supporting large movements during the acetyl transfer reaction, figure 5 is speculative. We have now edited Figure 5 in the revised version of the manuscript based on the observations we made in this study.

(3) The acetyl-CoA-bound state is described as the open-to-lumen state. Indeed, from Figure 1C, the lumen opening appears much larger than the cytosol opening. Is there any small tunnel that connects the substrate site to the cytosol? In other words, is this state accessible to both the lumen and the cytosol, albeit with a larger opening toward the lumen? This question arises because, in Figure S5, the tunnel calculated by MOLE seems to also connect to the cytosol.

Yes, it is likely that the ACOS is accessible via lumen and cytosol to varying degrees, as evidenced by MOLE prediction. However, binding of the bulky nucleoside head group of acetyl-CoA at ACOS blocks the cytosolic entrance in the confirmation discussed in this manuscript. MOLE prediction was performed on a structure devoid of acetyl-CoA, and it is possible that the protein doesn’t essentially undergo isomerization between open-to-lumen and open-to-cytosol confirmations during acetyl transfer. Likely, ACOS is always accessible from both the lumen and cytosol, but depending on the substrates or products bound, the accessibility could be limited to either the lysosomal lumen or cytosol. We have rewritten all the statements mentioning an open-to-lumen confirmation to reflect this argument.

(4) The authors state, "Interestingly, in most of the detergent conditions we tested, HGSNAT was predominantly dimeric (Fig S1C-H)," and also mention, "In all the detergents we tested, HGSNAT eluted as a dimer, a testament to the extensive side-chain interaction network." The dimerization is said to be mediated by a disulfide bond. I would be surprised if the detergents the authors tested could break a disulfide bond. Therefore, can this observation truly serve as a testament to an "extensive" side-chain interaction network?

We agree with the reviewer that detergents are unlikely to break a disulfide bond. To address this comment, we generated a C334A mutant of HGSNAT and extracted it from cells in 1% digitonin. It is still expressed as a dimer (Fig S8E). However, upon heating the detergent solubilized protein, we noticed that the FSEC peak for C334A shows a monomeric HGSNAT (Fig S8I and S8K). We hypothesize that in the absence of C334 disulfide, the extensive hydrophobic side-chain interaction network displayed in Figure 2C is responsible for maintaining the integrity of the dimer. Heating disturbs these non-disulfide interactions, thereby rendering the protein monomer.

(5) Apart from the cryo-EM structure, the article does not provide any other experimental evidence to support or explain a molecular mechanism. Due to the complete absence of functional assays, mutagenesis analysis, or other structures such as a ternary complex or an acetylated enzyme intermediate, the mechanistic model depicted in Figure 5 should be taken with caution. This uncertainty needs to be clearly described in the manuscript text. Performing additional mutagenesis experiments to test key hypotheses, or further discussing relevant data from the literature, would strengthen the manuscript.

We agree with the reviewer on the lack of supporting evidence for the mechanistic models proposed in Fig 5. They were made based on previously reported biochemical characterization of HGSNAT by Rome & Crain (1981), Rome et al. (1983), Miekle et al. (1995), and Fan et al. (2011). However, we agree with the reviewer that this schematic is not experimentally proven and is speculative at best. We have edited Figure 5 in the revised version of the manuscript. In addition, we have also performed mutagenesis analysis to study the stability of mutants (Fig S8) and performed LC-MS analysis to identify endogenously bound acetyl-CoA (Fig S9) to strengthen parts of the manuscript. We have discussed our findings in the results and modified the discussion according to these suggestions.

(6) It is discussed that H269 is an essential residue that participates in the acetylation reaction, possibly becoming acetylated during the process. However, there is no solid experimental evidence, e.g. mutagenesis analysis or structural analysis, in this or previous articles, that demonstrates this to be the case. Providing more information, ideally involving additional experimental work, would strengthen this aspect of the mechanism that is proposed. This would require establishing an in vitro assay, as described in 1.

H269, as a crucial catalytic residue, was suggested by monitoring the effect of chemical modifications of amino acids on acetylation of HGSNAT membranes by Bame, K. J. and Rome, L. H. (1986). We generated N258I and H269A mutants of HGSNAT and analyzed their stability. We noticed a greater destabilization in N258I compared to H269A (Fig S8). We believe this is because of the loss of ability to bind acetyl-CoA, as the TMs around a catalytic core of the protein in our cryo-EM structure were stabilized by interactions with acetyl-CoA. Recently, Xu et al. (2024, Nat Struct Mol Biol) suggested that they do not observe acetylated histidine in their structure. However, our structure and that reported by Xu et al. (2024) are obtained at cytosolic pH. Perhaps, acetylation of H269 occurs at acidic lysosomal pH. Extensive structural and catalytic investigation of HGSNAT at low pH is required to rule out H269 acetylation as a step in the HGSNAT catalyzed reaction.

(7) In the discussion part, the authors mention previous studies in which it was postulated that the catalytic reaction can be described by a random order mechanistic model or a Ping Pong Bi Bi model. However, the authors leave open the question of which of these mechanisms best describes the acetylation reaction. The structure presented here does not provide evidence that could support one mechanism or the other. The authors could explore if an in vitro experimental measurement of protein activity would provide any information in this regard.

We agree with the reviewer that a more detailed kinetic analysis is necessary to define the bisubstrate reaction mechanism of HGSNAT. All the existing structural data on two isoforms of HGSNAT is obtained at basic pH. As a result, the existing structures do not unambiguously demonstrate the bisusbtrate mechanism of HGSNAT. We believe low pH structural characterization and a detailed kinetic and structural characterization of HGSNAT in membrane mimetics like nanodiscs could provide more insights into the mechanism. However, these studies are a future undertaking and are not a part of this manuscript.

(8) Although the authors map the mutations leading to MPS IIIC on the structure and use FoldX software to predict the impact of these mutations on folding and fold stability, there is no experimental evidence to support FoldX's predictions. It would be ideal if an additional test for these predictions were included in the manuscript. The authors could follow the unfolding of purified mutants by SEC, FSEC, or changes in intrinsic fluorescence to assess protein stability.

As suggested here, we prepared HGSNAT MPSIIIC variants and tested their expression and stability (please see Fig S8). These results have been included in the revised version of the manuscript.

(9) Some sidechains that have quite strong sidechain density are missing atoms. I would be particularly careful with omitting sidechains that pack in the hydrophobic core, as this can tend to artificially reduce the clash score. Check F81, L62, P91 and V87, for example.

We have revisited the modeling of these regions and deposited new coordinates.

(10) W316 seems to have the wrong rotamer.

This has been corrected in the new coordinate file that has been released.

(11) N134 and N433 seem to have extra density. Are these known glycosylation sites?

As per Hrebicek M. et al., 2006 and Feldhammer M. et al., 2009, there are five predicted glycosylation sites: N66, N114, N134, N433, and N602. However, we see evidence for NAG density at N114, N134, and N433. These have now been modeled in the structure.

(12) At the C-terminal residue (Ile-635), the very C-terminal carboxylate is modeled pointing to a hydrophobic environment. It seems more likely to me that the Ile sidechain is packing here, with the C-terminal carboxylate facing the solvent.

Thank you for pointing this out. We have edited the orientation of the Ile sidechain accordingly.

Presentation and wording of results/methods:- Figure S3 legend "At places with missing density, the side chains were trimmed to C- alpha" - this is incorrect, I think the authors mean C-beta.

We have corrected this error in the revised version of the manuscript.

- Figure S3 legend - the authors refer to a gray mesh, where a transparent surface is displayed.

Thanks for pointing this error out. We have corrected this in the revised version.

- Some colloquial/vague wording in the main text a lot of sentences starting with "Interestingly, ...". Making the wording more specific would help the reader I think.

We have edited out ‘interestingly’ from the document and have re-written parts of the manuscript, per reviewers’ suggestion, for brevity.

- Figure S2 legend, "throughout the processing workflow the resolution of luminal domain was used as a guidepost" - it is not entirely clear to me what this means in this context, perhaps revise the wording?

We have rephrased this line in the revised draft of the manuscript.

- Figure S2 and methods, Local refinements of LD and TMD are mentioned, but not indicated on the processing workflow.

We have included a new Fig S2 & edited the legend, including these changes, per the reviewers’ suggestions.